# The bactericidal FabI inhibitor Debio 1453 clears antibiotic-resistant *Neisseria gonorrhoeae* infection in vivo

Vincent Gerusz [1] ✉, Pierre Regenass[1], Quentin Rousseau[1], Victor Moraine[1], Justine Dao[2], Xavier Lavé [2], Shampa Das[3], Josée Hue Perron [2], Laurence Fajas Descamps[2], Juan Bravo[2], Guennaëlle Dieppois[2], Nachum Kaplan[4], Matthew Lefebre[4], Deanna Altomari[4], Vladimir Romanov[4], Terry Finn[2], Pierre Daram[2], Francesca Bernardini[2], Michaël Gross[1], Robert Lysek[1], Aurélien Adam[1], Danig Pohin[1], Maurizio Maio[1], Vasileios Tatsis[5], Mihiro Sunose[5], Céline Ronin [6], Fabrice Ciesielski[6], Josefine Ahlstrand[7], Susanne Jacobsson[7], Magnus Unemo [7,8] & David R. Cameron [2] ✉

Gonorrhoea is a prevalent sexually transmitted infection caused by the bacterial pathogen *Neisseria gonorrhoeae*. *N. gonorrhoeae* has demonstrated a remarkable capacity to evolve antibiotic resistance, with emerging strains that show resistance to all standard treatment options. The development of new antibiotics for gonorrhoea, especially those with novel targets and no pre-existing resistance, is critical. One such untapped antibacterial target in *N. gonorrhoeae* is FabI, an enoyl-acyl carrier protein reductase enzyme that is essential for fatty acid biosynthesis in this pathogen. In the current report, structure-based drug design using novel *N. gonorrhoeae* FabI inhibitor co-crystals guides medicinal chemistry toward increasing potency in the sub-nanomolar range and drives the discovery of Debio 1453. Debio 1453 is optimized for activity against *N. gonorrhoeae* and is highly active in vitro against diverse *N. gonorrhoeae* isolates including those resistant to the last remaining treatment options. Additionally, the compound presents a low propensity for selection of mutants with reduced susceptibility. Debio 1453 is efficacious in vivo against *N. gonorrhoeae* isolates with clinically relevant multi-drug resistance phenotypes in a murine vaginal gonorrhoea infection model underscoring Debio 1453 as a promising candidate for the treatment of gonorrhoea.

The rise in antimicrobial resistance (AMR) for *Neisseria gonorrhoeae*, the bacterial pathogen responsible for the sexually transmitted infection gonorrhoea, is a significant global public health concern. If poorly treated, *N. gonorrhoeae* infection can lead to complications including pelvic inflammatory disease, ectopic pregnancy, infertility, and increased risk of acquisition and transmission of HIV and other sexually transmitted infections[1–4].

*N. gonorrhoeae* adapts rapidly to adverse environments; in the context of AMR, the gonococcus has evolved most of the classical physiological mechanisms for AMR, and this is at least partly attributable to its natural competence for transformation (i.e. the capacity to readily acquire genetic material horizontally) and genome plasticity[5]. The discovery and introduction of major antibiotic classes during the 20th century saw AMR spread in waves. More recently, increasing

azithromycin resistance rates and the spread of ceftriaxone-resistant strains, which is the only remaining first-line treatment in many countries, have further fuelled the threat of 'untreatable gonorrhoea'[1,5–8]. Controlling this threat is multimodal and involves enhanced surveillance, diagnosis, appropriate antibiotic stewardship, and the promotion of safe sexual practices, alongside the crucial development of new antibiotics with a distinct mechanism of action that are effective against circulating AMR clones.

Inhibition of FabI, which is an enoyl-acyl carrier protein (ACP) reductase that catalyses the essential rate-limiting step in fatty acid biosynthesis in some bacterial species, is an effective antimicrobial approach for various Gram-positive and Gram-negative bacterial pathogens[9–12]. The organisation of the bacterial fatty acid synthase type II (FASII) system is based on the activity of a series of mono-functional enzymes and is fundamentally different from the multifunctional FAS polypeptide responsible for fatty acid synthesis in mammals[13], thus providing good prospects for bacterial selectivity. Importantly, FabI is essential for the growth of *N. gonorrhoeae* and its chemical inhibition cannot be overcome by the addition of exogenous fatty acids in vitro[14,15], underscoring FabI as an attractive and yet untapped antibacterial target for the treatment of gonorrhoea. The most advanced FabI-inhibitor currently in clinical development is afabicin (Debio 1450), a prodrug of Debio 1452, that is potent against staphylococci and has demonstrated safety and preliminary efficacy for staphylococcal skin and skin structure infections in phase II clinical trials[16,17]. At high concentrations, Debio 1452 inhibits the growth of *N. gonorrhoeae* via inhibition of the *N. gonorrhoeae* FabI (*Ng*FabI) enzyme[15], which constitutes a promising starting point for lead optimisation against *N. gonorrhoeae*.

In this work, we describe the discovery and evaluation of Debio 1453, an optimised FabI-inhibitor potent against *N. gonorrhoeae* in vitro, and effective for the treatment of multidrug-resistant gonorrhoea in a murine model of infection.

## Results

### Development of *Ng*FabI inhibitors and the discovery of Debio 1453

A medicinal chemistry programme was undertaken, generating over 300 Debio 1452 analogues that were used to delineate Structure-Activity Relationships (SAR) to specifically increase compound potency against *Ng*FabI (Supplementary Table 1). The central scaffold based on a pyrido-enamide, along with a right-hand side amide constitutes the main pharmacophore that is crucial for activity (Fig. 1a). The remaining cycloalkyl lactam chain, as well as the left-hand side benzofuran, are amenable to modifications that could improve activity against *N. gonorrhoeae* and/or modulate absorption, distribution, metabolism and excretion (ADME) properties. Co-crystallisation studies revealed the binding mode of this chemical series within the *Ng*FabI catalytic pocket as a ternary complex involving the FabI active site, the NADH co-factor and the inhibitor (Supplementary Fig. 1). Such structures served the dual function of (i) reinforcing SAR from classical medicinal chemistry, and (ii) enabling structure-based drug design.

Key increases in potency were achieved by expanding the right-hand side lactam to a diazepanone, from which the extra nitrogen provided an additional interaction with the enzyme via H-bonding to the carbonyl of A199 (compound 1; IC50 6 nM; Fig. 1b and Fig. 2a). Various left-hand side modifications were assessed (e.g. compound 3, 4 and 5 in Supplementary Table 1) and demonstrated good activity against *Ng*FabI, however, the 2-branched benzofuran of compound 1 remained the most potent moiety. Inserting a stereochemically defined methyl group on this diazepanone resulted in supplemental Van der Waals interactions with the lipophilic environment of the active site (compound 2; IC50 2.4 nM; Fig. 1b and Fig. 2b) while its enantiomer was 5-fold less potent (compound 6; IC50 14 nM; Supplementary Table 1). The introduction of a hydroxyl group in a suitable orientation to this right-hand side reinforced the interaction with the

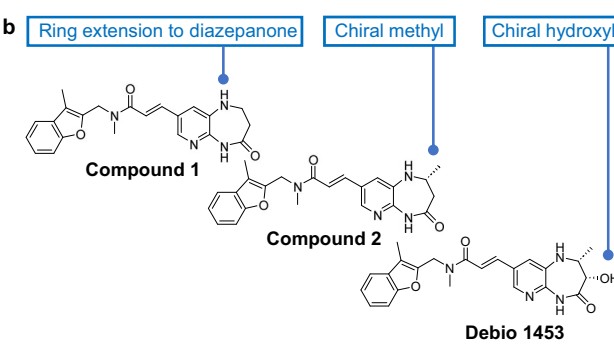

**a** Debio 1452

**Benzofuran**
- Mostly in lipophilic pocket
- Some modification permitted

**Lactam cycloalkyl chain**
- Directed to solvent
- Extensive latitude for modification

**Pyrido-enamide**
- Key pharmacophore
- Cannot be modified

**Amide**

**b** Ring extension to diazepanone | Chiral methyl | Chiral hydroxyl

Compound 1

Compound 2

Debio 1453

| | | | |
|---|---|---|---|
| *Ng*FabI IC50 | 6 nM | 2.4 nM | 0.6 nM |
| *Ng*MIC50 | 0.25 µg/ml | 0.125 µg/ml | 0.03 µg/ml |

**Fig. 1 | Discovery of Debio 1453. a** Afabicin desphosphono (Debio 1452) with structural characteristics governing FabI inhibition. **b** Representative key compounds in lead optimisation for *Neisseria gonorrhoeae* FabI (*Ng*FabI). *N. gonorrhoeae* minimum inhibitory concentration (MIC)50 values were determined for a screening panel of 14 isolates. IC, inhibitory concentration. Source data are provided as a Source Data file.

NADH co-factor via a conserved water bridging network (compound 7; IC50 1.9 nM; Supplementary Table 1; water bridging network similar to that shown in Fig. 2a). Again, the other stereoisomer was less potent (compound 8; IC50 15 nM; Supplementary Table 1). Finally, bringing together the stereochemically defined methyl and hydroxyl groups enhanced potency, resulting in Debio 1453 (IC50 0.6 nM; Fig. 1b and Fig. 2a). Improvements in *Ng*FabI potency were generally paralleled by increased antibacterial activity (Fig. 1b and Supplementary Table 1).

In addition to the binding features described above, Debio 1453 interacts with *Ng*FabI via a lipophilic T-shaped edge-to-face stacking in the left-hand side pocket between its benzofuran and Y159, a H-bond between its central carbonyl and the phenol of Y159, Van der Waals contacts between its right-hand side methyl group and the enzyme lipophilic environment, and a H-bond chelate between its right-hand side pyrido-amide and the backbone of A97 (Fig. 2). Interestingly, Debio 1453 also interacts with the NADH co-factor via Van der Waals contacts between its benzofuran methyl and the reduced form of nicotinamide, and via a H-bond between its central carbonyl and the ribose hydroxyl of NADH (Fig. 2). These multiple points of interaction between Debio 1453 and both the enzyme and its co-factor may account for its sub-nanomolar *Ng*FabI potency.

### Debio 1453 activity against *N. gonorrhoeae* and non-gonococcal *Neisseria spp.* in vitro

Debio 1453 activity was determined for the 2024 World Health Organisation *N. gonorrhoeae* reference strains[18] that include strains resistant to ceftriaxone, azithromycin, spectinomycin, tetracycline and/or ciprofloxacin (Debio 1453 MIC range 0.03–0.125 µg/mL; Table 1). Further, Debio 1453 activity was assessed against 100 consecutive, contemporary clinical *N. gonorrhoeae* isolates and compared with clinically relevant antibiotic comparators (Fig. 3). All isolates were

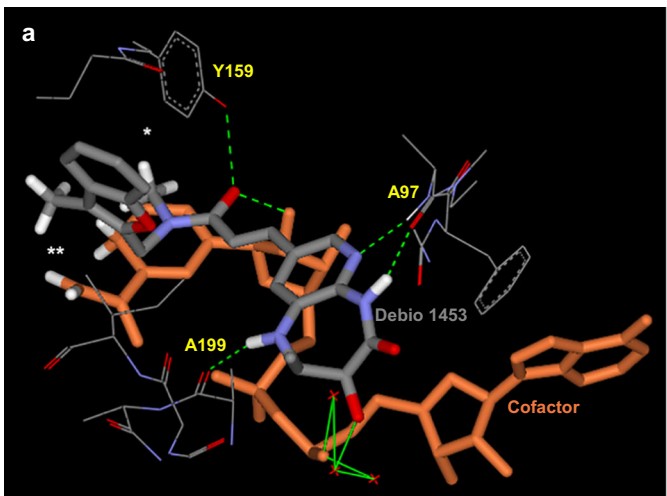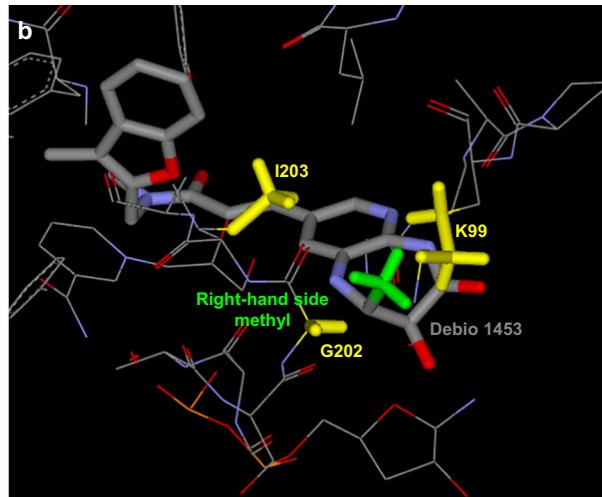

**Fig. 2 | Key structural interactions between Debio 1453 and *Ng*FabI. a** Main interactions of Debio 1453 (bold grey) with *Neisseria gonorrhoeae* FabI active site (key amino acids in light grey) and NADH co-factor (bold orange) from the co-crystallised ternary structure. H-bonds between Debio 1453 and FabI or NADH are represented by dashed green, while H-bonds between Debio 1453 and NADH via a conserved water network are in solid green. Debio 1453 is also interacting with the enzyme via T-stacking (*) and with the co-factor via Van der Waals hydrophobic interactions (**). **b** View of the right-hand side methyl group (bold green) of Debio 1453 (bold grey) in the *N. gonorrhoeae* FabI active site (amino acids and NADH in light grey) with surrounding 3 lipophilic amino acids (bold yellow) from the co-crystallised ternary structure.

**Table 1 | Minimum inhibitory concentration (MIC) values for Debio 1453 against the 2024 World Health Organisation (WHO) reference panel of *Neisseria gonorrhoeae* strains[18]**

| Strain | AMR phenotype (MIC, µg/mL)[a] | | | | | Debio 1453 MIC (µg/mL)[b] |
|---|---|---|---|---|---|---|
| | Ceftriaxone | Azithromycin | Ciprofloxacin | Spectinomycin | Tetracycline | |
| WHO F | S (<0.002) | S (0.25) | S (0.004) | S (16) | S (0.25) | 0.03 |
| WHO H | R (0.25) | S (0.25) | R (>32) | S (8) | R (4) | 0.06 |
| WHO K | S (0.06) | S (0.5) | R (>32) | S (16) | R (2) | 0.06 |
| WHO L | R (0.25) | S (1) | R (>32) | S (16) | R (2) | 0.03 |
| WHO M | S (0.016) | S (0.5) | R (2) | S (16) | R (2) | 0.06 |
| WHO O | S (0.032) | S (0.5) | S (0.008) | R (>1024) | R (2) | 0.06 |
| WHO P | S (0.004) | R (4) | S (0.004) | S (8) | R (1) | 0.125 |
| WHO Q | R (0.5) | R (<256) | R (>32) | S (8) | R (128) | 0.06 |
| WHO R | R (0.5) | S (0.5) | R (>32) | S (8) | R (4) | 0.06 |
| WHO S2 | S (<0.008) | R (2) | S (0.032) | S (16) | R (2) | 0.06 |
| WHO U | S (0.002) | R (4) | S (0.004) | S (8) | R (1) | 0.125 |
| WHO V | S (0.06) | R (>256) | R (>32) | S (16) | R (4) | 0.06 |
| WHO X | R (2) | S (0.5) | R (>32) | S (16) | R (2) | 0.06 |
| WHO Y | R (1) | S (1) | R (>32) | S (16) | R (4) | 0.06 |
| WHO Z | R (0.5) | S (1) | R (>32) | S (16) | R (4) | 0.06 |

*AMR* antimicrobial resistance, *R* resistant, *S* susceptible.
[a]AMR phenotypes (susceptibility categories) are according to EUCAST[76].
[b]Debio 1453 and ceftriaxone MICs were determined using agar dilution. Azithromycin, ciprofloxacin, spectinomycin, and tetracycline MICs were determined using Etest (bioMérieux, Marcy-Étoile, France).

susceptible to ceftriaxone and spectinomycin, whilst 2%, 40% and 46% percent of isolates were resistant to azithromycin, tetracycline and ciprofloxacin, respectively. The Debio 1453 MIC$_{50}$ and MIC$_{90}$ for the contemporary clinical panel were both 0.06 µg/mL (range 0.008 – 0.125 µg/mL).

Debio 1453 activity was next assessed for a panel of non-gonococcal *Neisseria* isolates, including 112 isolates representing 16 *Neisseria* species, including 15 commensal *Neisseria* species and *N. meningitidis* from various genogroups (Supplementary Fig. 2A). In general, those species with FabI sharing high amino acid identity with *Ng*FabI (>99%) were equally susceptible to Debio 1453 in vitro. Conversely, species coding for FabI with lower *Ng*FabI identity tended to

have higher MICs (Supplementary Fig. 2B). The propensity for horizontal acquisition of *fabI* alleles from non-gonococcal *Neisseria* species into *N. gonorrhoeae* was assessed using the Basic Local Alignment Search Tool (BLAST)[19]. No non-*Ng*FabI sequences were detected in the *N. gonorrhoeae* taxid (Taxonomy ID 485 at NCBI), which included 38,623 publicly available complete or draft genomes (Supplementary Fig. 2B, note: *N. polysaccharea* and *N. lactamica* FabI share 100% identity with *Ng*FabI).

## Debio 1453 in vitro time-kill kinetics
Debio 1453 displayed rapid bactericidal killing against both antibiotic-susceptible and extensively drug-resistant (XDR) *N. gonorrhoeae*

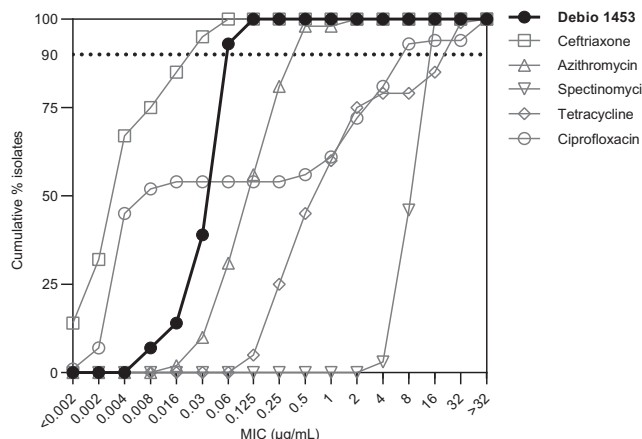

**Fig. 3 | Cumulative minimum inhibitory concentration (MIC) distributions for Debio 1453 and comparator antibiotics for 100 clinical *Neisseria gonorrhoeae* isolates collected from Sweden in 2023.** Relevant EUCAST resistance breakpoints or epidemiological cutoff (ECOFF) for *N. gonorrhoeae*: ceftriaxone, >0.125 µg/mL; azithromycin, 1 µg/mL (ECOFF); spectinomycin, >64 µg/mL; tetracycline, >0.5 µg/mL; and ciprofloxacin, >0.06 µg/mL. The dotted line indicates 90% of cumulative isolates (MIC90). Source data are provided as a Source Data file.

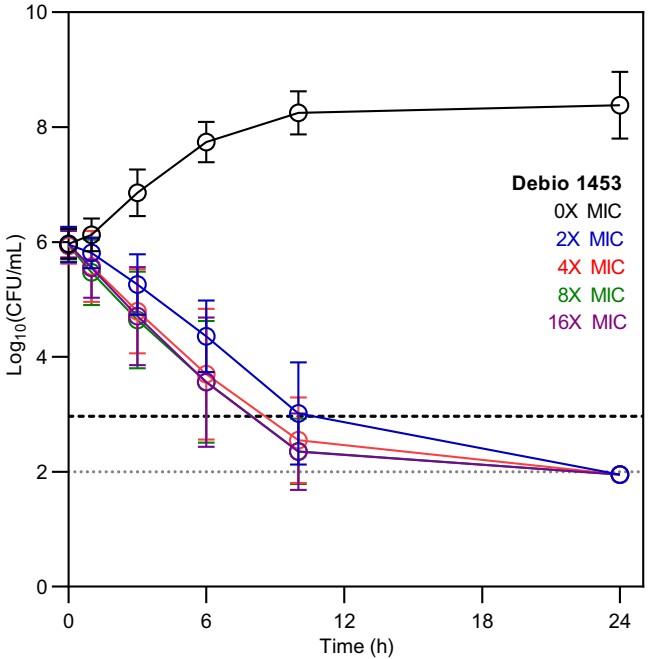

**Fig. 4 | Bactericidal activity of Debio 1453 against *Neisseria gonorrhoeae*.** Combined in vitro time-kill data for 10 *N. gonorrhoeae* strains at increasing multiples of Debio 1453 minimum inhibitory concentration (MIC). Data are the mean +/− standard deviation (n = 10). Time-kill curves for each individual strain are presented in Supplementary Fig. 3. Source data are provided as a Source Data file.

achieving ≥3 log10 reductions in viable CFU counts within 24 h in time-kill assays at all of the concentrations tested, including as low as 2× MIC (collective data for 10 isolates is presented in Fig. 4, data for each individual isolate tested is presented in Supplementary Fig. 3). The mean time to reach bactericidal activity across the ten isolates was similar for all of the concentrations tested (10 h at 2× MIC, 8.5 h at 4× MIC, 8 h at 8× MIC and 8 h at 16× MIC) suggesting time-dependent killing in vitro. Further, *N. gonorrhoeae* infection of the human genital mucosa involves invasion and persistence within epithelial cells[20,21] and as such, the capacity for Debio 1453 to kill intracellular *N. gonorrhoeae*

was assessed within cultured HeLa229 human cervix carcinoma cells. Debio 1453 was effective against both internalised antibiotic-susceptible (American Type Culture Collection [ATCC] 49226; Fig. 5a) and XDR *N. gonorrhoeae* (WHO X, Fig. 5b), eradicating the bacteria to the limit of detection within 24 h for each isolate tested (≥3.1 log10CFU/mL and ≥2.8 log10CFU/mL reductions, respectively). Intracellular activity for Debio 1453 was similar to the control antibiotic, azithromycin, for both strains tested (≥3.3 log10CFU/mL and ≥2.8 log10CFU/mL reductions for ATCC 49226 and WHO X, respectively; Fig. 5).

## Debio 1453 selection for mutants with reduced in vitro susceptibility

Spontaneous single-step mutation rates for Debio 1453 were determined using previously described methods that were adapted for *N. gonorrhoeae*[22]. Mutant frequencies using an inoculum of ~10^8 CFU were determined at three concentrations (4×, 8×, 16× MIC) for strain ATCC 700825 in biological triplicate, and ciprofloxacin was included for control (Table 2). The frequency of mutant selection was low. For two of the replicates, no viable colonies were detected, producing frequencies of mutant generation that were below the limit of quantification. For the final replicate, the frequency of mutant generation at 4× and 8× the MIC was 7.06 × 10^−9, whilst no viable colonies were detected at 16x the MIC. The mutant frequency at 16× MIC for ATCC 700825 ranged from <6.38 × 10^−9 to <9.38 × 10^−9. Ciprofloxacin frequencies of resistance were similar to those previously published[23]. Three additional *N. gonorrhoeae* isolates were assessed (6926, 6804, WHO Z) at a higher inoculum (~10^9 CFU) (Table 2). Viable colonies were selected at 4× MIC, with frequencies ranging from 1.06 × 10^−9 to 2.81 × 10^−10. No colonies were selected at 8× or 16× the MIC (frequency values ranging from <5.28 × 10^−10 to <2.81 × 10^−10). Two independent clones of strain 6804 that showed a stable 4-fold increase in MIC (from 0.06 to 0.25 µg/mL) were selected for *fabI* gene and promoter sequencing, and each revealed the same G771T nucleotide change in the *fabI* coding sequence, resulting in a predicted L257F amino acid substitution in FabI, thus further validating FabI as the target for Debio 1453.

## Development of phosphate prodrug Debio 1453 P for in vivo assessments

Debio 1453 has limited aqueous solubility (29 µg/mL at pH 7.4 after 24 h in PBS with 1% DMSO), and as such, development of a prodrug was initiated to improve solubility. Phosphate ester prodrugs are known to be cleaved by in vivo phosphatases to release the corresponding hydroxyl-containing drug, while the presence of the dianionic phosphate promoiety usually increases aqueous solubility[24–26]. Importantly, phosphate esters are hydrolysed at similar rates in different preclinical species by alkaline phosphatases[27]. Encouraged by the increased solubility of the prodrug fosfluconazole and its in vivo conversion by alkaline phosphatases to the antifungal drug fluconazole despite a sterically hindered phosphate[28], this concept was applied to Debio 1453. The phosphate prodrug (Debio 1453 P) had drastically improved aqueous solubility (>100 mg/mL), facilitating a water-soluble formulation designed to release the active compound when exposed to host phosphatases in vivo.

Conversion of Debio 1453 P to Debio 1453 by intestinal cells was demonstrated using human colon carcinoma (Caco-2) cell monolayers grown on transwell plates. In control experiments without Caco-2 cells, Debio 1453 P (initial nominal concentration of 100 µM) demonstrated high permeability through transwells from apical (A) to basolateral (B; A→B) compartments of 31.8 × 10^−6 cm/s with 97% total recovery following 120 min of incubation, and conversion to Debio 1453 was negligible. In the presence of Caco-2 monolayers, the mean recovery of Debio 1453 P decreased to 43.3%, and the recovery of Debio 1453 increased to 44.2% (with combined total recovery irrespective of compartment of 87.5%), which suggested prodrug conversion

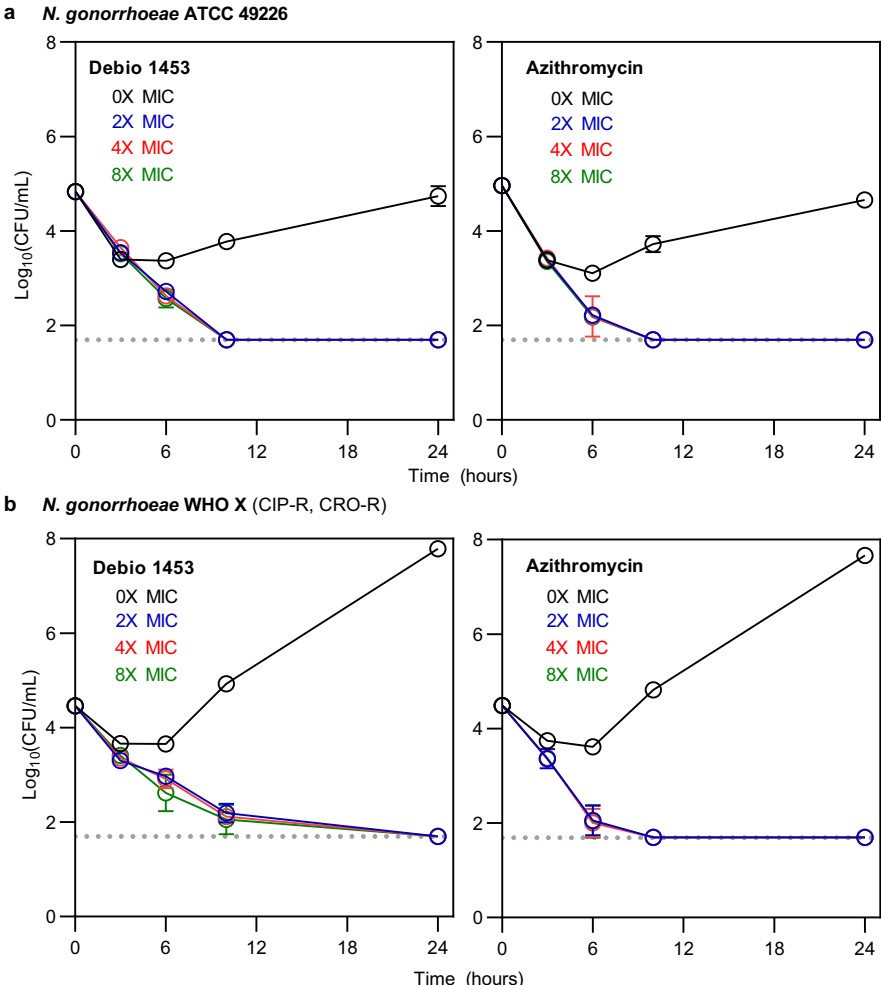

**Fig. 5 | Intracellular killing of *Neisseria gonorrhoeae* in cultured HeLa229 human cervix carcinoma cells. a** Intracellular killing of *N. gonorrhoeae* strain ATCC 49226 at increasing multiples of Debio 1453 or azithromycin minimum inhibitory concentration (MIC). **b** Intracellular killing of *N. gonorrhoeae* strain WHO X at increasing multiples of Debio 1453 or azithromycin MIC. Data are the mean of biological triplicates +/− standard deviation. Dotted grey lines indicate the limit of detection. CFU colony forming unit, CIP-R ciprofloxacin-resistant, CRO-R ceftriaxone-resistant. Source data are provided as a Source Data file.

**Table 2 | Frequency of selection for mutants with elevated minimum inhibitory concentration (MIC) for three strains of *Neisseria gonorrhoeae* exposed to increasing concentrations of ciprofloxacin or Debio 1453**

| Antibiotic | Fold MIC | | *N. gonorrhoeae* strain | | | |
|---|---|---|---|---|---|---|
| | | ATCC 700825[a] | 6926 | 6804 (CIP-R) | WHO Z (CIP-R, CRO-R) |
| Ciprofloxacin | 4× | $6.38 \times 10^{-9}$ | $1.50 \times 10^{-9}$ | NT | NT |
| | 8× | $<6.38 \times 10^{-9}$ | $1.88 \times 10^{-9}$ | NT | NT |
| | 16× | $<6.38 \times 10^{-9}$ | $3.75 \times 10^{-10}$ | NT | NT |
| Debio 1453 | 4× | $<6.38 \times 10^{-9}$ $<9.38 \times 10^{-9}$ $7.06 \times 10^{-9}$ | $3.85 \times 10^{-10}$ | $1.06 \times 10^{-9}$ | $2.81 \times 10^{-10}$ |
| | 8× | $<6.38 \times 10^{-9}$ $<9.38 \times 10^{-9}$ $7.06 \times 10^{-9}$ | $<3.85 \times 10^{-10}$ | $<5.28 \times 10^{-10}$ | $<2.81 \times 10^{-10}$ |
| | 16× | $<6.38 \times 10^{-9}$ $<9.38 \times 10^{-9}$ $<7.06 \times 10^{-9}$ | $<3.85 \times 10^{-10}$ | $<5.28 \times 10^{-10}$ | $<2.81 \times 10^{-10}$ |

Instances where no mutant colonies were observed are denoted as less than (<) one colony divided by the starting inoculum. NT, not tested as both 6804 and WHO Z are ciprofloxacin-resistant. CIP-R, ciprofloxacin-resistant; CRO-R, ceftriaxone-resistant. Relevant Debio 1453 MICs for this assay were 0.016 µg/mL for ATCC 700825, 0.03 µg/mL for 6926, 0.06 µg/mL for 6804 and 0.125 µg/mL for WHO Z. Source data are provided as a Source Data file.
[a]ATCC 700825 was performed in biological triplicate using an inoculum of ~10[8] CFU; the remaining strains were performed once using an inoculum of ~10[9] CFU.

mediated by Caco-2 cells. Apparent permeability ($P_{app}$) of Debio 1453 P was negligible (below detection limit in the basolateral compartment). In contrast, $P_{app}$ for Debio 1453 was $7.93 \times 10^{-6}$ cm/s from A → B and $30.5 \times 10^{-6}$ cm/s from B → A (mean total recovery of 95%), indicative of medium-to-high in vitro permeability. Metabolic stability was next assessed in human hepatocytes. Intrinsic clearance ($Cl_{int}$) of Debio 1453 P was low (<0.44 µL/min/million cells) and low-to-intermediate for Debio 1453 (1.37–4.85 µL/min/million cells). Together, early ADME studies highlighted prodrug conversion to the active moiety Debio 1453 and promising potential for oral bioavailability and metabolic stability.

## Debio 1453 demonstrates low potential for toxicity in vitro

Potential for toxicity of Debio 1453 P and Debio 1453 was assessed using several in vitro test systems. Debio 1453 P was non-mutagenic in the Ames assay in both the presence and absence of S9 metabolism at the highest concentrations tested (Supplementary Table 2); Debio 1453 could not be assessed using this assay due to its intrinsic antimicrobial activity. Instead, Debio 1453 was assessed in a mouse lymphoma assay and was non-mutagenic at the highest dose level tested when assessing induction of 5 trifluorothymidine in vitro (Supplementary Table 3). Both compounds tested negative in the in vitro micronucleus test, showing no clastogenic or aneugenic activity in lymphocytes irrespective of the presence or absence of a rat liver metabolic activation system (Supplementary Tables 4 and 5). Cytotoxicity of Debio 1453 was assessed in human hepatocytes (HepG2); at the highest concentration tested (30 µM), no cytotoxicity or negative impact on cell viability was observed (Supplementary Table 6).

## Debio 1453 efficacy in *N. gonorrhoeae*-infected mice

Systemic in vivo exposures of Debio 1453 following a single oral dose of Debio 1453 P (80 mg/kg Debio 1453 equivalent) were assessed using estradiol-treated ovariectomised mice infected intravaginally with *N. gonorrhoeae* strain ATCC 700825. Debio 1453 P was quantified in plasma shortly after dosing ($C_{max}$ 37.2 ng/mL, $t_{max}$ 0.083 h; Supplementary Fig. 4), albeit at concentrations more than 100-fold below Debio 1453 levels, indicating a rapid in vivo conversion of the prodrug to the active moiety. The free exposure of Debio 1453 in plasma (considering the measured mouse plasma protein binding of 93.7%) was above the Debio 1453 MIC90 for *N. gonorrhoeae* (Fig. 6), providing encouraging prospects for in vivo efficacy.

The in vivo efficacy of Debio 1453 (dosed as Debio 1453 P) was assessed for four *N. gonorrhoeae* challenge strains using the murine vaginal infection model. ATCC 700825 is susceptible to ceftriaxone and azithromycin, whereas AR Bank0179-15 and AR Bank0181-17 were ceftriaxone-susceptible and azithromycin-resistant, and WHO X-07 was ceftriaxone-resistant and azithromycin-susceptible. Each strain colonised the genital tract as evidenced by >1log10CFU/mL increases in vaginal lavage fluid from the start of treatment to 24 h and sustained bacterial loads at 48 h. Ceftriaxone was included for control and was effective against each of the ceftriaxone-susceptible strains (Fig. 7a–c). At equivalent doses for efficacy against ceftriaxone-susceptible strains, ceftriaxone was ineffective against the ceftriaxone-resistant strain WHO X-07, however, efficacy was demonstrated when higher doses were used (Fig. 7d). Debio 1453 was efficacious against each of the strains, producing clear dose responses, achieving ~3log10CFU/mL reductions as compared to the start of treatment (0 h) and eradicating *N. gonorrhoeae* to the limit of detection by 48 h for at least one dose level.

To evaluate the translatability of the pharmacodynamic targets defined above, the *N. gonorrhoeae* vaginal model was benchmarked using data generated for ceftriaxone. The pharmacokinetic-pharmacodynamic (PK-PD) driver for ceftriaxone efficacy is time of free concentration above MIC ($fT > MIC$). PK data from healthy mice for single doses of 0.5, 1 and 5 mg/kg were simulated from a previous

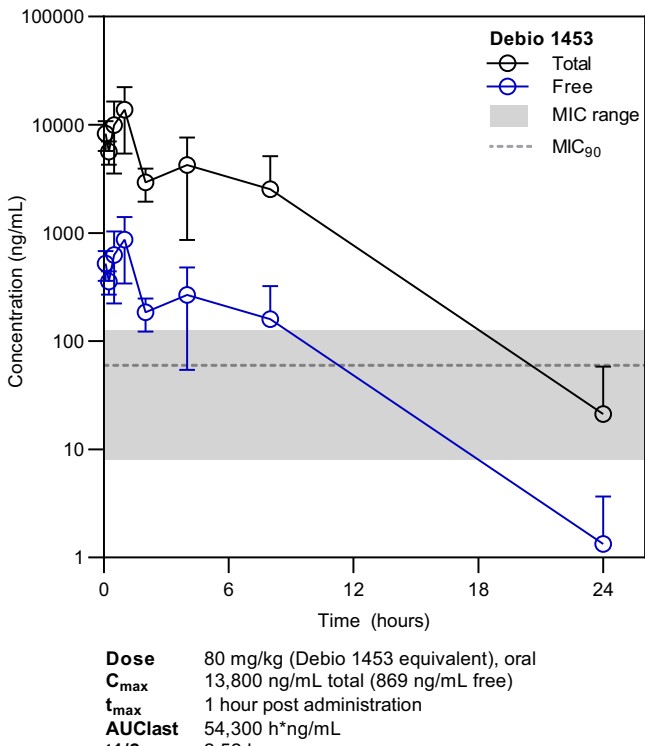

**Dose** 80 mg/kg (Debio 1453 equivalent), oral
**$C_{max}$** 13,800 ng/mL total (869 ng/mL free)
**$t_{max}$** 1 hour post administration
**AUClast** 54,300 h*ng/mL
**t1/2** 2.52 hours

**Fig. 6 | Plasma exposure of Debio 1453 dosed as Debio 1453 P in a *Neisseria gonorrhoeae* murine vaginal infection model.** Total (black) and free (blue, accounting for mouse plasma protein binding of 93.7%) Debio 1453 concentrations are the mean from three animals per timepoint, with standard deviation. Error bars below zero cannot be displayed due to the logarithmic scale for the *x*-axis. The lower limit for quantification of total Debio 1453 was 10 ng/mL. The minimum inhibitory concentration (MIC) range for Debio 1453 is highlighted in grey, and the MIC inhibiting 90% of *N. gonorrhoeae* isolates (MIC90) is represented by the grey dotted line. Pharmacokinetic parameters were determined using Phoenix WinNonlin by non-compartmental analysis. The area under the curve (AUC)last is for total Debio 1453. $C_{max}$ maximum concentration, $t_{max}$ time taken to reach $C_{max}$, t1/2 half-life. Source data are provided as a Source Data file.

study published by Connolly et al. [29] (Supplementary Fig. 5A), and used to relate $fT > MIC$ with the change in log10CFU/mL at 24 h (Supplementary Fig. 5B). A dose with simulated $fT > MIC$ of 2 h was not efficacious in the model (1.9 log10CFU/mL increase), whereas doses with simulated $fT > MIC$ of 12 to 18 h, ≥22 to 24 h, and >24 h resulted in 2.0, 2.9 and 3.3 log10CFU/mL reductions, respectively.

## Debio 1453 efficacy in surrogate *Staphylococcus aureus*-infected neutropenic mice

The *Staphylococcus aureus* neutropenic murine thigh infection model, which has been used extensively to provide informative antibiotic efficacy data for human translation[30,31], was next used to assess the in vivo activity of Debio 1453. *S. aureus* ATCC 29213 performed well in vehicle controls (3.1 and 3.5 log10CFU/mL increases from 0 to 24 and 48 h, respectively; Fig. 8). Linezolid 100 mg/kg three times daily was included as a control, and was predicted to produce exposures higher than the human clinical equivalent (120 mg/kg twice daily dosing in mice emulates 600 mg twice daily human dosing)[32]. Debio 1453 (dosed as Debio 1453 P) achieved >1 log10CFU/mL reductions in thighs at 24 h and >2 log10CFU reductions at 48 h, displaying maximal efficacy that was comparable to supratherapeutic linezolid treatment (Fig. 8).

## Discussion

Overcoming antibiotic-resistant *N. gonorrhoeae* is likely best achieved by exploiting novel targets and/or acting via a distinct mechanism of

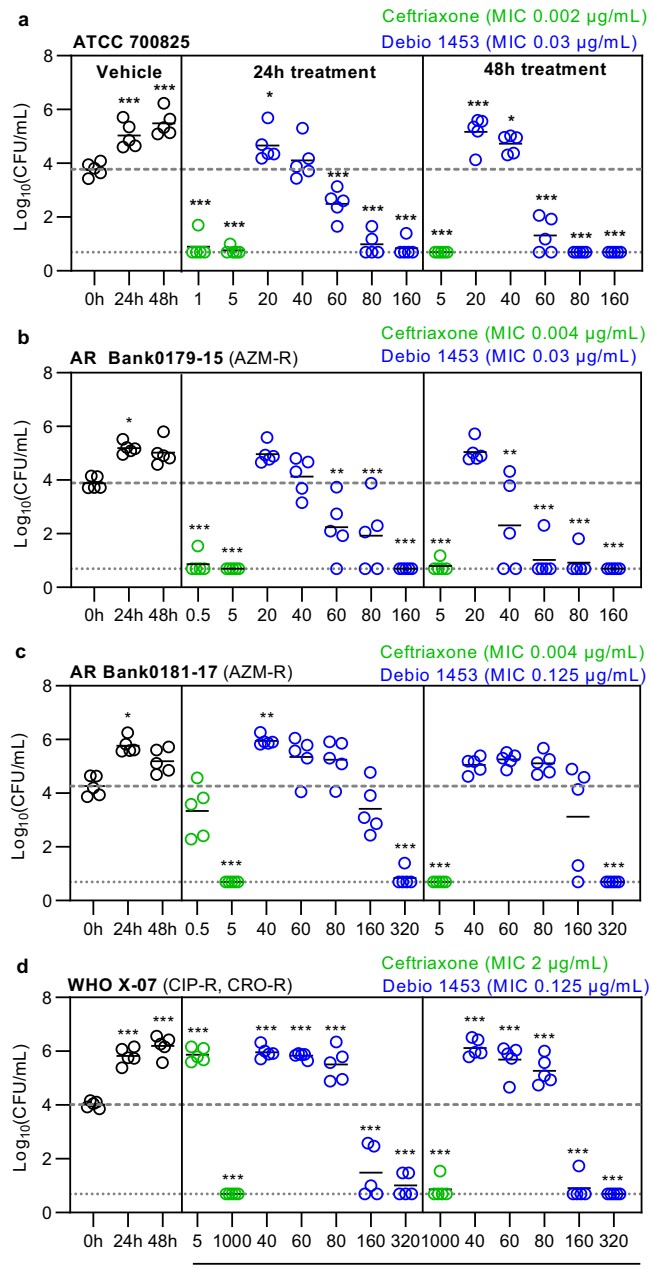

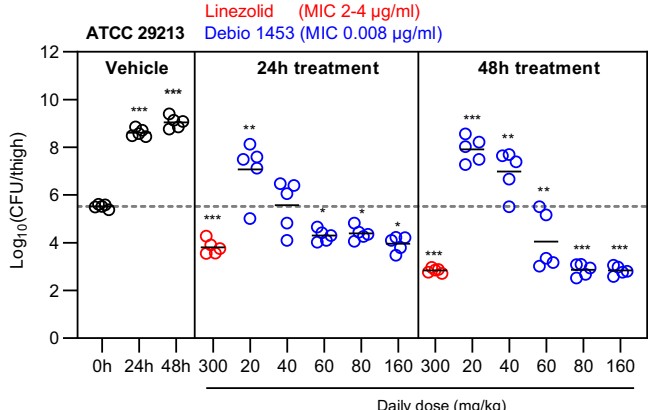

**Fig. 8 | In vivo efficacy of Debio 1453 in a *Staphylococcus aureus* neutropenic murine thigh infection model.** The infective strain, *S. aureus* ATCC 29213, was inoculated 2 h before the start of treatment (0 h). Thighs were harvested at 24 h and 48 h for colony-forming unit (CFU) determinations. Data are summarised by the mean for each group (black horizontal bars, *n* = 5 animals per group). Debio 1453 was dosed as Debio 1453 P; doses are Debio 1453 equivalents. Debio 1453 P was dosed orally, twice daily. Linezolid was dosed orally three times daily. The dotted grey lines depict the mean CFU/mL at the start of treatment. Statistically significant differences compared to 0 h were determined using one-way ANOVA with Dunnett's multiple comparisons test. *$P < 0.05$, **$P < 0.01$, ***$P < 0.001$. MIC, minimum inhibitory concentration. Source data, including statistical test output with exact *P*-values, are provided as a Source Data file.

**Fig. 7 | In vivo efficacy of Debio 1453 for the treatment of *Neisseria gonorrhoeae* in a murine vaginal infection model.** Bacterial counts (colony forming units, CFU) per millilitre of vaginal lavage fluid for mice infected with *N. gonorrhoeae* strains **a** ATCC 700825, **b** AR Bank0179-15, **c** AR Bank0181-17, and **d** WHO X-07, treated with either ceftriaxone (single intraperitoneal dose) or Debio 1453 P (dosed orally twice daily; doses are Debio 1453 equivalent values). Minimum inhibitory concentrations (MICs) were determined using agar dilution and are the mode of multiple measures. Dotted lines indicate the limit of detection, dashed grey lines show mean CFU/mL at the beginning of treatment. ATCC 700825 is resistant to streptomycin, facilitating use in the model, whereas AR Bank0179-15, AR Bank0181-17 and WHO X-07 were engineered for streptomycin resistance from progenitors AR Bank0179, AR Bank0181 and WHO X, respectively. Data for each group is summarised using the mean (black horizontal bars, *n* = 5 animals per group). Statistically significant differences compared to 0 h were determined using one-way ANOVA with Dunnett's multiple comparisons test. *$P < 0.05$, **$P < 0.01$, ***$P < 0.001$. AZM-R azithromycin-resistant, CIP-R ciprofloxacin-resistant, CRO-R ceftriaxone-resistant. Source data, including statistical test output, are provided as a Source Data file.

action. Leveraging on successes against problematic AMR bacteria, including methicillin-resistant *S. aureus*[16], Debio 1453 is a FabI-inhibitor tailored for activity against *N. gonorrhoeae*. One common challenge for antibiotics that target a single enzyme is the potential for rapid single-step resistance emergence (i.e. via acquisition of a single mutation within the target)[33,34]. Yet, Debio 1453 displays a very low propensity for single-step resistance emergence (Table 2) with mutations to the target enzyme only producing modest increases in the MIC of Debio 1453. Structural insights generated with co-crystal structures have delineated a two-component binding mode for this series of FabI inhibitors, whereby, in addition to interacting with the FabI active site, part of the surface of the inhibitor also associates with the NADH co-factor (Fig. 2a). This two-component binding mode has generally been observed for pyrido-enamide scaffold-based FabI inhibitors in *S. aureus* or *Escherichia coli* FabI active sites[22,35]. Since the co-factor is non-mutable, medicinal chemistry efforts therefore focused on maximising interactions with NADH in the hope of minimising mutational resistance. This was achieved by Van der Waals hydrophobic interactions with the reduced amido-pyridine of NADH, and a hydrogen bond between the central carbonyl of Debio 1453 and the ribose hydroxyl of NADH. The *Ng*FabI co-structures described here achieved resolution of 1.3 Å, higher than those reported previously for *S. aureus* FabI (1.8 Å)[21] or *E. coli* FabI (1.5 to 2.7 Å)[35], which provided sufficient detail to leverage the interactions of the right-hand side hydroxyl substituent with a precisely defined conserved water network binding to the co-factor NADH phosphate (Fig. 2a), thereby reinforcing non-mutable interactions and increasing potency. As a result, these multiple interactions position Debio 1453 in such a way that a significant part of its molecular surface is facing the co-factor and thus cannot be affected by mutational changes. Due to the high number of key interactions in the active site, it is also expected that multiple mutations would be required at the core of the catalytic pocket to significantly destabilise this complex, which would likely come at a cost for enzyme functionality and

overall fitness, similar to what was postulated previously for afabicin[36]. In support of this postulation, the only amino acid substitution that we observed in the current study that was associated with a Debio 1453 MIC increase (NgFabI L257F) occurred at a residue that is not within the active site (Supplementary Fig. 6). On the contrary, this residue lays at the interface of another NgFabI subunit only present in the tetrameric form. Therefore, this mutation might rather affect the enzyme oligomeric transition[37,38] by disrupting interface interactions between monomeric subunits as already observed for resistant *S. aureus* FabI[39].

Non-gonococcal *Neisseria* species can serve as a reservoir of AMR determinants for *N. gonorrhoeae*[40–43]. Non-gonococcal *Neisseria* isolates with MIC values outside of the range for *N. gonorrhoeae* were identified in this study, and differences in MIC appeared to relate to the degree of FabI amino acid sequence identity, as was expected from the structure-dependent binding mode of the inhibitor. Other explanations for variance were not assessed (i.e. differences in active sites sequence identity, drug uptake, efflux, metabolism), and remain a topic for future work. Horizontal acquisition of *fabI* from non-gonococcal *Neisseria* species was not observed using public genome databases (*n* = 38,623 complete or draft *N. gonorrhoeae* genomes), albeit in the absence of a direct selection pressure. Future studies are warranted to determine if *fabI* alleles from non-gonococcal *Neisseria* species can be acquired by *N. gonorrhoeae* upon Debio 1453 exposure in co-culture.

The estradiol-treated mouse model of vaginal colonisation is an emerging tool for the assessment of antimicrobial efficacy[29,44–49]. In the current study, using estradiol-treated ovariectomised mice, four different *N. gonorrhoeae* strains were assessed and each successfully colonised the genital tract for at least 50 h post-inoculation, providing an adequate therapeutic window to assess antibiotic efficacy. Furthermore, quantifiable dose responses were observed, allowing for the future definition of PK-PD targets for efficacy. The translatability of these findings for clinical application was assessed by first benchmarking the model using the last remaining first-line gonorrhoea treatment, ceftriaxone. Historically, $fT > $ MICs roughly equating to 7–10 h were used clinically based on data for penicillin[50,51]. Targets were revised to combat strains with elevated ceftriaxone MIC, with optimal efficacy found to require $fT > $ MIC of 20–24 h[50]. In the *N. gonorrhoeae* murine vaginal infection model, $fT > $ MIC simulations for ceftriaxone of 12 to 18 h, ≥22 to 24 h-, and >24 h produced 2.0, 2.9 and 3.3 $\log_{10}$CFU/mL reductions in vaginal lavage fluid, respectively, providing support for the clinical translatability of 2- and 3 $\log_{10}$CFU/mL reduction targets using the model, each of which were achieved by Debio 1453 treatment. The findings were further validated using the 'gold standard' neutropenic thigh *S. aureus* model, whereby Debio 1453 treatment achieved efficacy comparable to that of linezolid administered at levels predicted to be supratherapeutic when translated to humans.

Overall, the FabI target for Debio 1453 was validated by sub-nanomolar NgFabI $IC_{50}$ values, NgFabI-Debio 1453 co-crystal structures, and the identification of FabI mutations following controlled in vitro exposure. Debio 1453 displayed (i) in vivo efficacy in mice against isolates resistant to clinically relevant antibiotic classes, including the last remaining first-line treatment ceftriaxone and azithromycin, (ii) bactericidal activity against planktonic *N. gonorrhoeae* cells as well as internalised/intracellular bacterial cells and (iii) a low propensity for selection of mutations that increase the Debio 1453 MIC. Each of these key properties is aligned to the preferred target product profile (TPP) for the treatment of uncomplicated gonorrhoea, as described by the WHO and the Global Antibiotic Research and Development Partnership (GARDP)[52]. The promising findings from this report support the continued development of Debio 1453 for the treatment of infections caused by *N. gonorrhoeae*.

## Methods

### Chemical synthesis

NMR Spectra were acquired on a Bruker Avance III spectrometer at 400 MHz using residual undeuterated solvent as reference. LC/MS analyses were performed on a Waters X-Select CSH C18, 2.5 μm, 4.6 × 30 mm column eluting with a gradient of 0.1% formic acid in MeCN in 0.1% formic acid in water. The gradient from 5–95% 0.1% formic acid in MeCN occurs between 0.00–3.00 min at 2.5 mL/min with a flush from 3.01–3.5 min at 4.5 mL/min. A column re-equilibration to 5% MeCN is performed from 3.60–4.00 min at 2.5 mL/min. UV spectra of the eluted peaks were measured using an Agilent 1260 Infinity or Agilent 1200 VWD at 254 nm. Mass spectra were measured using an Agilent 6120 or Agilent 1956 MSD running with positive/negative switching or an Agilent 6100 MSD running in either positive or negative mode. Synthesis of Compound 1, Compound 2, Debio 1453 and Debio 1453 P is detailed below. Synthesis of additional compounds from Supplemental Table 1 is described in the Supplementary Methods.

**Compound 1.** Prepared as described previously (referred to as compound 11 d)[53].

**Compound 2.** A flask was charged with (R)-8-bromo-2-methyl-2,3-dihydro-1H-pyrido[2,3-b][1,4]diazepin-4(5H)-one (prepared as described previously[54] compound 213, 80 mg, 0.31 mmol), N-methyl-N-((3-methylbenzofuran-2-yl)methyl)acrylamide (prepared as described previously[54] compound 9, 70 mg, 0.31 mmol), tetrabutylammonium chloride hydrate (9.2 mg, 0.03 mmol) and Pd-116 (16 mg, 0.03 mmol) and the flask was evacuated and backfilled with nitrogen three times. 1,4-Dioxane (2.5 mL) and N,N-diisopropylethylamine (0.11 mL) were added, and the reaction mixture was heated to 90 °C for 2 h. Water was added, and the aqueous mixture was extracted with dichloromethane. The combined organic layers were dried by passing through a phase separation cartridge and concentrated in a vacuum. The product was purified by chromatography on silica gel (0–10% methanol/dichloromethane) to generate compound **2** as a yellow solid (68 mg, 52%). Rt 1.89 min $m/z$ 405 [M + H]$^+$ (ES + ). $^1$H NMR (DMSO-d$_6$, 278 K): $\delta$, ppm 9.88 (s, 1H), 8.10 & 8.07 (rotamers, s, 1H), 7.57 (d, $J$ = 8.0 Hz, 1H), 7.50 (d, $J$ = 8.0 Hz, 1H), 7.48–7.41 (m, 2H), 7.35 & 7.08 (rotamers, d, $J$ = 16 Hz, 1H), 7.29 (dt, $J$ = 8.0 Hz, 1.6 Hz, 1H), 7.25 (dt, $J$ = 8.0 Hz, 1.6 Hz, 1H), 5.70-5.66 (m, 1H), 4.96 & 4.80 (rotamers, s, 2H), 3.86–3.76 (m, 1H), 3.18 & 2.95 (rotamers, s, 3H), 2.57 (dd, $J$ = 14 Hz, 2.5 Hz, 1H), 2.39 (dd, $J$ = 14 Hz, 8.0 Hz, 1H), 2.28 & 2.27 (rotamers, s, 3H), 1.21 (d, $J$ = 6.0 Hz, 3H).

$^1$H NMR (DMSO-d$_6$, 363 K): $\delta$, ppm 9.32 (s, 1H), 8.02 (s, 1H), 7.55 (d, $J$ = 8.0 Hz, 1H), 7.53–7.38 (m, 3H), 7.30–7.22 (m, 2H), 7.12 (d, $J$ = 16 Hz, 1H), 5.50–5.44 (m, 1H), 4.84 (s, 2H), 3.86–3.76 (m, 1H), 3.10 (s, 3H), 2.61 (dd, $J$ = 14 Hz, 2.5 Hz, 1H), 2.45 (dd, $J$ = 14 Hz, 8.0 Hz, 1H), 2.28 (s, 3H), 1.25 (d, $J$ = 6.0 Hz, 3H).

**Debio 1453.** A reaction vial was charged with N-methyl-N-((3-methylbenzofuran-2-yl)methyl)acrylamide (prepared as described previously[54], compound 9, 42 mg, 0.18 mmol), (2 R,3S)-8-bromo-3-hydroxy-2-methyl-2,3-dihydro-1H-pyrido[2,3-b][1,4]diazepin-4(5H)-one (prepared as described previously[54] compound 149, 50 mg, 0.18 mmol), tetrabutylammonium chloride (5.0 mg, 0.02 mmol) and [P(tBu)3]Pd(crotyl)Cl (Pd-162) (7.0 mg, 0.02 mmol) and the vial was evacuated and backfilled with nitrogen three times. 1,4-Dioxane (5.0 mL) and N-cyclohexyl-N-methylcyclohexanamine (79 μL, 0.37 mmol) were added, and the reaction mixture was heated to 80 °C and stirred for 16 h. The reaction was allowed to cool to room temperature, the solvent was removed under vacuum, and the solid was washed with isohexane. The crude material was then purified by column chromatography on silica gel (0–3% MeOH/DCM), which afforded a pale yellow solid that was partially dissolved in acetonitrile. Water

was added until precipitation occurred. The precipitate was collected by filtration and dried under azeotropic conditions with acetonitrile to afford Debio 1453 as a pale yellow solid (35 mg, 45%). Absolute configuration was confirmed by co-crystallisation (*vide infra*). Rt 1.83 min $m/z$ 421 $[M+H]^+$ (ES +). $^1$H NMR (500 MHz, DMSO-d6, 363 K): $\delta$, ppm 9.80 (s, 1H), 7.98 (d, $J$ = 1.9 Hz, 1H), 7.59–7.52 (m, 1H), 7.49–7.37 (m, 3H), 7.31–7.22 (m, 2H), 7.12 (d, $J$ = 15.7 Hz, 1H), 5.91 (d, $J$ = 5.7 Hz, 1H), 4.84 (s, 2H), 4.76 (d, $J$ = 4.7 Hz, 1H), 4.22 (dd, $J$ = 4.7 Hz, 3.4 Hz, 1H), 3.82–3.71 (m, 1H), 3.10 (s, 3H), 2.27 (s, 3H), 1.12 (d, $J$ = 6.5 Hz, 3H).

**Debio 1453 P**. To a suspension of Debio 1453 (1.01 g, 2.40 mmol, 1 eq.) in dry dichloromethane (24.4 mL) at 0 °C was added bis(2-cyanoethyl)-diisopropylphosphoramidite (1.88 mL, 7.21 mmol) and 1H-tetrazole (16.0 mL, 7.21 mmol, 0.45 M in acetonitrile). The mixture was stirred at 0 °C for 3 h. Then, a solution of 0.2M iodine in water/pyridine/tetrahydrofuran (1/19/80) was added at 0 °C until the persistence of iodine colouration. The mixture was stirred at 0 °C for 5 min. A solution of sodium thiosulfate (20% w/w in water) was added until the removal of iodide colouration. Water was added, and the organic phase was extracted with ethyl acetate. After drying and concentration of the organic phase, the residue was purified on silica gel eluting with 0–10% methanol in dichloromethane and evaporated resulting in bis(2-cyanoethyl) [(2 R,3S)-2-methyl-8-[(E)-3-[methyl-[(3-methylbenzofuran-2-yl)methyl]amino]-3-oxo-prop-1-enyl]-4-oxo-1,2,3,5-tetrahydropyrido[2,3-b][1,4]diazepin-3-yl] phosphate (1.35 g, 2.23 mmol, 92.6%) as a yellow solid. To a solution of this solid (3.45 g generated via multiple iterations of the above, 5.69 mmol, 1 eq.) in dry acetonitrile (42 mL) at 0 °C were added trimethylsilyl (1E)-2,2,2-trifluoro-N-trimethylsilyl-ethanimidate (5.91 mL, 28.4 mmol) and 2-tert-butyl-1,1,3,3-tetramethylguanidine (7.28 mL, 28.4 mmol). After 10 min, formic acid (2.15 mL, 56.9 mmol) was added, and the mixture was evaporated. Purification by reverse phase C18 column yielded a yellow powder (2.62 g) after lyophilisation, which was dissolved in water (10 mL) and triethylamine (0.29 mL, 2.15 mmol). The solution was stirred at room temperature for two minutes. Sodium 2-ethylhexanoate (2.49 g, 14.6 mmol) was added, and the reaction mixture was stirred for two minutes. The reaction mixture was diluted with acetonitrile (100 mL). The precipitate was filtered and dissolved again in water three times, then finally dissolved in water and lyophilised to afford compound Debio 1453 P (1.72 g, 3.16 mmol, 56% for 2 steps) as a disodium salt. Rt 1.22 min $m/z$ 501.3 $[M+H]^+$ (ES +). $^1$H NMR (400 MHz, D2O): $\delta$ 7.94–7.86 (m, 1H), 7.47–6.77 (m, 7H), 4.71–4.53 (m, 2H), 4.23–4.12 (m, 1H), 3.05 (s, 1.7H), 2.97 (s, 1.3H), 2.16–2.09 (m, 3H), 1.31–1.21 (m, 3H). $^{31}$P (161 MHz, D2O) $\delta$ 3.4.

## Solubility determination
**Kinetic solubility of Debio 1453**. In a Multiscreen solubility filter plate (Millipore AG, Switzerland), 2 μL of a 10 mM stock solution in DMSO of test compounds was added to 198 μL of phosphate buffer (PBS pH 7.4). The filter plate was then incubated under shaking (orbital shaker, 1200 rpm) at 25 °C for 24 h. After incubation, the plate was filtered using a Multiscreen vacuum Manifold and collected in a 96-well Teflon plate. 100 μL of the filtrate was then transferred into a Greiner 96-well plate and diluted with 100 μL of acetonitrile before UV analysis. The quantification was performed using a reference solution of the compounds to test, prepared with 2 μL of DMSO stock solution diluted in 99 μL of PBS and 100 μL of Acetonitrile (1% DMSO final concentration).

**Solubility of Debio 1453 P**. 100 mg of product was weighed into a 1 mL flask. Water for injection was then added to the flask and shaken. Solubilisation was fully reached after 3 s.

## Co-crystallisation
*Ng*FabI protein was concentrated to 23 mg/mL and diluted to 20 mg/mL by the addition of NADH and three molar equivalents of Debio 1453. Co-crystals grew at 22 °C in 100 mM bicine, pH 9 and 2.4 M

ammonium sulfate. Drops were performed using the sitting drop vapour-diffusion method in 96-well plates (300 nL protein solution + 300 nL reservoir) with a Mosquito robot (SPT Labtech). Co-crystals were directly flash-frozen in liquid nitrogen. Data collection was performed on the PROXIMA-2 beamline at SOLEIL synchrotron (Saint-Aubin, France). Data were processed with XDS[55], autoPROC (GlobalPhasing)[56] and the CCP4 software package[57]. Structure was solved by molecular replacement with Phaser[58] using a published *Acinetobacter baumannii* FabI structure in complex with Debio 1452 and NADH as a search model (pdb: 6AHE)[59]. Model building and improvement were conducted by iterative cycles of automated and manual building with Coot[60], and refinement with Refmac[61]. Quality of the model was monitored with Rampage[62] (crystallographic and refinement data, Supplemental Table 7).

## *Ng*FabI inhibitory assays
FabI activity was monitored by measuring NADH consumption[9]. Reaction mixtures consisting of 100 mM Tris-HCl (pH 7.2), 100 mM ammonium acetate, 0.05% Pluronic F-68, 25 μM crotonyl-ACP, 50 pM recombinant *Ng*FabI protein, 7.5% DMSO and test compounds ranging from 0.00017 to 10 μM, were incubated for 10 min at 30 °C to facilitate inhibitor binding. To initiate reactions, 50 μM NADH was added, and absorbance was monitored at 340 nM every 12 s for a total of 40 min using a Spectramax Plus 384 plate reader (Molecular Devices) at a controlled temperature of 30 °C. Data were analysed using the XLfit Microsoft Excel plugin (version 5.5). $IC_{50}$ values were determined using a logistic sigmoid curve-fitting of the inhibitor dose response curves.

## Bacterial growth conditions
*N. gonorrhoeae* was routinely grown on chocolate agar plates (GC medium [BD Difco], supplemented with 1% haemoglobin from bovine blood [Sigma-Aldrich] and 1% (v/v) BBL IsoVitaleX Enrichment [BD]) and incubated overnight at 35–36 °C in a humid 5% $CO_2$–enriched atmosphere unless otherwise stated.

## MIC determinations and time-kill assays
Ceftriaxone and Debio 1453 MIC values were determined for *N. gonorrhoeae* using agar dilution methodology according to CLSI guidelines using the DMSO direct method for insoluble compounds[63,64]. MICs of azithromycin, ciprofloxacin, spectinomycin, and tetracycline were determined by Etest (bioMérieux, Marcy-Étoile, France) on GC agar base (GC Medium Base agar; BD Diagnostics, Sparks, MD, USA) supplemented with 1% IsoVitalex (BD Diagnostics). The 2024 WHO *N. gonorrhoeae* reference strains were used for quality control[18]. Only whole MIC doubling dilutions are reported. Clinical breakpoints or the epidemiological cutoff (ECOFF, to indicate azithromycin resistance) from EUCAST (v14.0, https://www.eucast.org/clinical_breakpoints) were used to classify isolates as antimicrobial susceptible or resistant. To generate baseline MIC values for liquid-based assays, broth microdilution methodology was used according to CLSI guidelines[63], with substitution of Cation-Adjusted Mueller-Hinton Broth for Columbia Broth (BD Difco) supplemented with 1% IsoVitaleX Enrichment (BD). Two times the standard inoculum defined by CLSI was used to establish consistent growth in untreated control samples for some strains. In vitro time-kill assays were performed according to CLSI guidelines[65] using Columbia Broth (BD Difco) supplemented with 1% IsoVitaleX Enrichment (BD). Viable bacterial counts were generated by plating 10-fold serially diluted samples on GC agar plates containing 1% IsoVitaleX Enrichment (BD). Debio 1453 MIC values for non-gonococcal *Neisseria* and *S. aureus* were determined using agar dilution and broth dilution, respectively, each according to CLSI guidelines[63].

## In silico comparison of FabI across *Neisseria* species
FabI amino acid sequences were retrieved from the NCBI protein repository and aligned using CLUSTAL omega[66]. Phylogenetic analysis was performed using Interactive Tree of Life (iTOL) v7.0[67].

## Intracellular killing

Intracellular killing was assessed within HeLa229 cells (European Collection of Authenticated Cell Cultures, London, UK) that were seeded at ~3 × 10^5 cells/mL and grown to 70–80% confluence in RPMI-glutamine media supplemented with 10% (v/v) heat-inactivated fetal bovine serum (FBS, Gibco, MA, USA). Cells were incubated at 37 °C with 5% $CO_2$ for 24 h, then washed with serum-free RPMI-1640 prior to infection. *N. gonorrhoeae* was scraped from agar plates and resuspended in Dulbecco's Phosphate Buffered Saline (DPBS, Sigma-Aldrich), pelleted, then resuspended in RPMI with 2% FBS. Bacterial suspensions were used to infect HeLa229 cells at a multiplicity of infection of 100:1. Plates were briefly centrifuged (500 × g for 5 min), then incubated for one hour. This incubation and all subsequent incubations were performed at 37 °C with 5% $CO_2$. Non-adherent cells were removed by washing with serum-free RPMI, and the remaining extracellular bacteria were killed via resuspension in RPMI-glutamine supplemented with 2% FBS and 200 µg/mL gentamycin and incubation for one hour. Cells were washed three times with serum-free RPMI, then resuspended in RPMI-glutamine medium supplemented with 2% FBS and Debio 1453 at various test concentrations, and then incubated. Azithromycin-treated samples were included for control/comparison. At appropriate time points, cells were washed three times with DBPS, then lysed with 1% saponin for 15 min. Samples were diluted and plated onto chocolate II agar for viable bacteria quantification. Intracellular killing assays were performed at Microbiologics, Kalamazoo, MI, USA, using a method adapted from previous studies[68,69].

## In vitro selection of mutants with reduced Debio 1453 susceptibility

*N. gonorrhoeae* strains were grown overnight on chocolate agar plates, as described above. Bacteria were spread onto test plates consisting of GC agar, 1% IsoVitaleX and either Debio 1453 or ciprofloxacin at either 4×, 8×, or 16× the respective compound MIC. Plates were incubated at 35 ± 2 °C in a humid 5% $CO_2$–enriched atmosphere for 48–72 h. Colonies that grew on selection plates were patched onto GC agar, 1% IsoVitaleX plates, with and without Debio 1453 at the same concentration used for selection, then incubated overnight to confirm clone viability and the stability of the phenotype with reduced Debio 1453 susceptibility. Frequencies were generated by dividing the number of stable colonies with reduced Debio 1453 susceptibility by the inoculated CFU.

## DNA sequencing

Genomic DNA was extracted from single colonies using the PureLink Genomic DNA Mini Kit (ThermoFisher) as per the manufacturer's instructions. The *N. gonorrhoeae fabI* gene including 160 upstream of the start codon was amplified using OneTAQ Hotstart according to manufacturer's instructions with the following primers (5′–3′); F5 CATCTGATGCCTTAAACCGTATTTG and R1 CAAGTCGGTACAAAGGCAATCG. Sanger sequencing of amplicons was performed at The Centre for Applied Genomics, Toronto, Canada.

## Caco-2 permeability and in vitro hepatic clearance determinations

The capacity for compounds to permeate Caco-2 intestinal epithelial cells was determined for Debio 1453 and Debio 1453 P[70,71]. Briefly, permeation from apical to basolateral compartment and basolateral to apical compartment of CacoReady-monolayers in 24-transwell plates (Readycell, Barcelona, Spain) was assessed with pH 7.4 on each side for Debio 1453, and with pH 6.5 on the apical side, and 7.4 on the basolateral side for Debio 1453 P. Incubations were performed at 37 °C, and sampling from both sides was performed at time zero and after 120 min. Various controls were assessed in parallel, including atenolol (low permeability) and diclofenac (high permeability).

Pooled human hepatocytes from 10-donor males (BioIVT, West Sussex, UK; MX008001) or 10-donor females (BioIVT, West Sussex, UK;

FX008001) were incubated with test compounds (or vehicle) and the compound concentration was measured over time. Samples were incubated at 37 °C and concentrations were assessed after 0, 20-, 40-, 60- and 240-min. Verapamil was included as a control.

For Caco-2 permeability and in vitro hepatic clearance assessments, Debio 1453 and Debio 1453 P (as appropriate) were quantified using LC-MS/MS. Chromatographic separation was achieved using an Acquity UPLC (Waters) equipped with a Phenomenex EVO C18 column (2.1 mm × 50 mm, 1.7 µm) with a flow rate of 0.5 mL/min. The mobile phase consisted of 5 mM ammonium formate in water, 0.2% (v/v) ammonia (A) and acetonitrile (B). The UPLC system was coupled to a Thermo Q-Exactive Orbitrap mass spectrometer (Thermo) that was used to acquire mass spectra with Thermo XCalibur (version 4.1.31.9). The instrument operated in positive ionisation mode (ESI + ) over a $m/z$ range of 80–1000 ($n$ = 1 measure per sample). Data were extracted from chromatograms using calculated monoisotopic accurate masses for protonated molecular ions within 5 ppm windows.

## Genotoxicity assessments

Debio 1453 and Debio 1453 P were assayed for the propensity to induce micronuclei in human lymphocytes following in vitro treatment with and without liver S9 fraction from rats[72]. Ames assays were performed using four strains of *Salmonella typhimurium* (TA1535, TA1537, TA98 and TA100) and a single strain of *E. coli* (WRP2 *uvrA*) in the presence and absence of S9 tissue homogenate[73]. Dose levels (µg/plate) up to 4000 were assessed for WP2 *uvrA* and TA98, 500 for TA1535, 250 for TA1537 and 62.5 for TA100. Mutagenic activity was further assessed for Debio 1453 by quantifying the induction of 5 trifluorothymidine-resistant mutants in mouse lymphoma L5178Y TK+/− cells (ATCC CRL 9518) after in vitro treatment, in the absence and presence of S9 metabolic activation, using the fluctuation method[74].

## Cytotoxicity assessment

HepG2 cells (HB-8065; ATCC) cultured in DMEM (Gibco) with 5% fetal bovine serum (heat-inactivated, Sigma-Aldrich) and 1% penicillin-streptomycin (ThermoFisher), were seeded in 96-well plates (3.0 × 10^4 per well) and incubated at 37 °C and 5% $CO_2$. Cells were treated with Debio 1453 (0.3, 1, 3, 10, 30 µM) or vehicle (1% DMSO) and incubated for a further 24 h (37 °C, 5% $CO_2$). Each treatment was assessed in triplicate. Cytotoxicity was determined using CytoTox-ONE™ Homogenous Membrane Integrity Assay kits (Promega), and cell viability was determined using CellTiter-Glo® Luminescent Cell Viability Assay kits (Promega), each according to the manufacturer's specifications.

## Plasma protein binding determination

Mouse plasma samples fortified with cOmplete™ EDTA-free Protease Inhibitor Cocktail (Roche, 1 tablet per 7 mL of plasma) were spiked with Debio 1453 (1 – 30 µM). Triplicate samples were pre-incubated at 37 °C for 15 min prior to dialysis using an HTDialysis device for 6 h at 37 °C in 5% $CO_2$. Debio 1453 in samples taken from each chamber was quantified using LC-MS/MS. Chromatographic separation was achieved using a Nexera 30 series LC system (Shimadzu) equipped with a Phenomenex Kinetex XB-C18 column (2.1 × 50 mm, 2.6 µm) with a flow rate of 0.6 mL/min. The mobile phase consisted of 10 mM ammonium formate, 0.2 % (v/v) ammonium hydroxide (A) and acetonitrile (B). The LC system was coupled to an API-4500 mass spectrometer (Sciex) that was used to acquire mass spectra with Sciex Analyst software (version 1.7.2). The instrument operated in positive ionisation mode (Turbo ionspray, $n$ = 1 measure per sample). Quantification was achieved using an isotopically labelled internal standard and Multiple Reaction Monitoring (MRM) with transition 421.1/145.1.

## *Neisseria gonorrhoeae* murine vaginal infection model

The *N. gonorrhoeae* female mouse vaginal model[45] was performed in accordance with the Guide for Care and Use of Laboratory Animals[75].

Animal rooms were maintained at a temperature range of 20–24 °C and humidity between 30% and 70%, with 12 h light/dark cycles. Ovariectomized BALB/c female mice (5–6 weeks, BioLASCO Taiwan Co., Ltd.) were treated with estradiol (0.23 mg/mouse) two days before infection and on the day of infection. Ovariectomized mice were used to avoid staging mice before treating them with estradiol[44]. Beginning two days prior to infection and maintained until the end of the study, animals were treated twice daily with streptomycin (1.2 mg/mouse) and vancomycin (0.6 mg/mouse) each via intraperitoneal (IP) injection and received trimethoprim sulfate in the drinking water (0.4 mg/mL) to control vaginal microflora. Animals were anesthetised with pentobarbital (80 mg/kg), the vagina was rinsed with 50 mM HEPES (pH 7.4), and then inoculated with *N. gonorrhoeae* in pre-warmed PBS (0.02 mL/mouse containing ~1 × 10⁵ CFU).

For pharmacokinetic assessments, at 2 h post-infection with ATCC 700825, Debio 1453 P formulated in 5% dextrose was administered as a single oral dose of 80 mg/kg (Debio 1453 equivalent) via oral gavage. Animals were euthanised via $CO_2$ asphyxiation, and blood was collected by cardiac puncture ($n = 3$ animals per timepoint). Blood was drawn into tubes coated with $K_2$EDTA, mixed gently, then centrifuged at 2500 × $g$ for 15 min at 4 °C. Plasma was mixed with human plasma with protease inhibitor (cOmplete™ EDTA-free Protease Inhibitor Cocktail, Roche) at a ratio of 1:1.5 and then frozen immediately. Debio 1453 P and Debio 1453 were quantified using LC-MS/MS. Chromatographic separation was achieved using an Acquity UPLC equipped with a BEH C18 column (2.1 × 50 mm, 1.7 μm; Waters) and a flow rate of 0.6 mL/min. The mobile phase consisted of 5 mM ammonium formate in water, 0.2 % (v/v) ammonium hydroxide (A) and acetonitrile (B). The UPLC system was coupled to a QTRAP 6500 mass spectrometer (Sciex) that was used to acquire mass spectra with Sciex Analyst software (version 1.6.2). The instrument operated in positive ionisation mode (TurboIon Spray, $n = 1$ measure per sample). Each analyte was quantified using a dedicated isotopic labelled internal standard and in Multiple Reaction Monitoring (MRM), using transition 421.1/363.00 for Debio 1453, and 501.0/145.1 for Debio 1453 P. The lower limit for quantification was 10 ng/mL. Pharmacokinetic analysis was performed using Phoenix 64 WinNonlin (version 8.4.0.6172) using extravascular models and the Linear Up Log Down calculation, sparse sampling method.

For efficacy assessments, at 2 h post-infection, animals began either Debio 1453 P (formulated in 5% dextrose) treatment via oral gavage, which was repeated every 12 h for either 24 or 48 h. Control groups received either a single IP injection of ceftriaxone or 5% dextrose (vehicle) twice daily. Animals were sacrificed via $CO_2$ asphyxiation at either 2 h (baseline), 26 and/or 50 h post-infection. Vaginal lavage was performed twice with 200 μL GC broth containing 0.05% saponin to recover bacteria. Bacterial burden in the lavage fluids was determined by performing 10-fold serial dilutions and plating on chocolate agar plates. Four strains were assessed using the model; ATCC 700825 is naturally streptomycin-resistant, whereas AR Bank0179-15, AR Bank0181-17 and WHO X-07 are derivatives of AR Bank0179, AR Bank0181 and WHO X, respectively, that were engineered for streptomycin resistance for use in the model.

### *Staphylococcus aureus* neutropenic murine thigh infection model

The *S. aureus* neutropenic murine thigh infection model was performed in accordance with the Guide for Care and Use of Laboratory Animals[75]. Animal rooms were maintained at a temperature range of 20–24 °C and humidity between 30% and 70%, with 12 h light/dark cycles. Female ICR mice (BioLASCO Taiwan Co., Ltd.) were rendered neutropenic via IP injection of cyclophosphamide four days before infection (150 mg/kg) and again one day before infection (100 mg/kg). Animals were anaesthetised with isoflurane (3–5%), then inoculated intramuscularly with *S. aureus* ATCC 29213 (~1 × 10⁵ CFU/mouse).

Animals received Debio 1453 P formulated in 5% dextrose via oral gavage twice daily for either 24 or 48 h. Control groups received either linezolid three times per day or 5% dextrose (vehicle) twice daily, each via oral gavage. Animals were sacrificed via $CO_2$ asphyxiation at either 2 h (baseline), 26 and/or 50 h post-infection. Thigh tissues were harvested and homogenised in 3 mL sterile PBS (pH 7.4). Bacterial burdens were determined by performing 10-fold serial dilutions and plating on nutrient agar plates.

### Ceftriaxone PK simulation

Ceftriaxone concentration time-profiles in uninfected mice dosed with 0.25 and 1.5 mg/kg from a previous study published by Connolly et al.[29] were digitised using WebPlotDigitizer (version 4.7). Data were used to simulate PK profiles at doses of 0.5, 1 and 5 mg/kg using non-parametric superposition using Phoenix WinNonlin (version 8.3.4.295).

### Statistical analysis

For in vivo efficacy experiments, significant differences between treatment groups and the 0-h group were assessed using one-way ANOVA with Dunnett's multiple comparisons test (GraphPad Prism version 9.3.1).

### Ethics statement

Experiments involving animals were performed with ethical approval from the Institute for Animal Care and Use Committee at Pharmacology Discovery Services Taiwan, Ltd.

### Reporting summary

Further information on research design is available in the Nature Portfolio Reporting Summary linked to this article.

## Data availability

All data supporting the findings of this study are available within the Source Data File. Structural data have been deposited and are available in the Protein Data Bank (PDB) under accession code 9S5X. Source data are provided with this paper.

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

## Acknowledgements

CARB-X funding for this research is supported by federal funds from the U.S. Department of Health and Human Services (HHS); Administration for Strategic Preparedness and Response; Biomedical Advanced Research and Development Authority; UK Department of Health and Social Care (GAMRIF); under agreement number 75A50122C00028, and by awards from Wellcome (WT224842) and Germany's Federal Ministry of Education and Research (BMBF). The content of this manuscript is solely the responsibility of the authors and does not necessarily represent the official views of CARB-X or its funders. This work utilised NIAID's suite of preclinical services for preclinical models of infectious diseases (Contract No. HHSN272201700020I/75N93023F00002). All in vivo animal models were performed at Pharmacology Discovery Services, Taiwan. We acknowledge the efforts of Camille Kowalski, Daniel Biasse and Caroline Mathon, and the staff of the Translational Laboratory at Debiopharm. We thank Zrinka Ivezic Schoenfeld for her insights that improved the manuscript. We also acknowledge the contribution of the teams at Jubilant Biosys, Eurofins Cerep, France; European Research Biology Centre, France and Italy; Admescope, Finland, Noida, India; Sygnature Discovery, Nottingham, UK; Evotec International GmbH, France; and Micromyx/Microbiologics, Kalamazoo, MI, USA.

## Author contributions

V.G., P.R., Q.R., V.M., T.F., M.G., R.L., A.A., D.P., M.M., V.T., and M.S. contributed to medicinal chemistry design, compound synthesis, analytics and formulation. J.B., G.D., N.K., M.L., D.A., V.R., P.D., F.B., J.A., S.J., M.U., and D.R.C. evaluated antibacterial properties, enzymology and/or microbiology. L.F.D. and J.B. were responsible for ADME. J.H.P. was responsible for toxicology. C.R. and F.C. performed crystallographic studies. J.D., L.F.D., X.L., S.D., M.U., and D.R.C. contributed to in vivo efficacy assessments. V.G. and D.R.C. wrote the manuscript with input from all authors.

## Competing interests

V.G., V.T., M.S., J.B., and T.F. are listed as inventors on the patent application WO2020099341 "Antibiotic compounds, methods of manufacturing the same, pharmaceutical compositions containing the same and uses thereof" which covers compound 2, Debio 1453 and Debio 1453P listed in this manuscript. V.G., P.R., Q.R., V.M., M.G., R.L., A.A., D.P., M.M., were each employees of Debiopharm Research and Manufacturing SA. J.D., X.L., J.H.P., L.F.D., J.B., G.D., T.F., P.D., F.B., and D.R.C. were each employees of Debiopharm International SA. S.D. and N.K. were paid consultants of Debiopharm International SA. The remaining authors declare no competing interests.

## Additional information

[1]Debiopharm Research and Manufacturing SA, Martigny, Switzerland. [2]Debiopharm International SA, Lausanne, Switzerland. [3]Antimicrobial Pharmacodynamics and Therapeutics, Department of Pharmacology, University of Liverpool, Liverpool Health Partners, Liverpool, UK. [4]Nobelex Biotech, Inc., Toronto, ON, Canada. [5]Sygnature Discovery, Biocity, Nottingham, UK. [6]Novalix, Strasbourg, France. [7]WHO Collaborating Centre for Gonorrhoea and Other STIs, National Reference Laboratory for STIs, Department of Laboratory Medicine, Faculty of Medicine and Health, Örebro University, Örebro, Sweden. [8]Institute for Global Health, University College London (UCL), London, UK. ✉e-mail: vincent.gerusz@debiopharm.com; david.cameron@debiopharm.com

