## [Transparent Peer Review file · Nature Communications]

The bactericidal FabI inhibitor Debio 1453 clears antibiotic-resistant *Neisseria gonorrhoeae* infection in vivo

Corresponding Author: Dr Vincent Gerusz

Version 1:

Reviewer comments:

Reviewer #1

(Remarks to the Author)

Please see the attached document.

Reviewer #2

(Remarks to the Author)

This is a well-written and thorough study on the optimisation of the FabI-inhibitor, Dehbio 1452.

I do note that prior studies have used other estradiol-treated mouse models of vaginal colonisation to report efficacy of antimicrobials and indeed have reported PK/Pd for ceftriaxone (See A. Jerse studies) (line 220). It would be worth commenting on why you felt that ovariectomies were necessary in an estradiol-treatment model.

I felt that adding a commentary on whether Dehbio would result in escape mutants in commensal *Neisseria* spp. would be worth adding here. It is well known that escape mutations appear preferentially in the non-clinical species and so showing that they are not under selective pressure would add to the case for continued development of this promising lead compound.

Reviewer #3

(Remarks to the Author)

General:

N. gonorrhoea strains have developed resistance to nearly all medications and novel treatments are sorely needed.

Strengths:

The authors describe a new antimicrobial compound that inhibits an essential function involved in fatty acid synthesis in bacteria. The compound displayed antibacterial activity against a broad range of *N. gonorrhoea* strains; exhibited efficacy in a murine vaginal gonorrhoea model and *S. aureus* neutropenic murine thigh model; and showed very low bacterial resistance.

Weakness:

Toxicity is a principal concern of antimicrobials. Enthusiasm for the paper would be strengthened if toxicity to mammalian cells or animals was directly addressed in the manuscript.

Specific

1. Table 1. Define what is depicted by shading
2. Fig. S2. State data are a mean of X determinations +/- SD
3. Fig. 3. Define what is indicated by dotted line
4. Fig. 4B/C / Fig. S3. Add the type of cultured mammalian cells used for intracellular killing assays to legend; consider merging the figures in manuscript so controls are shown with the experiment.
5. Table 2. State data are a mean of X determinations +/- SD...or a statement that values given are within a +/- percentage
6. Fig. 5. Define horizontal bar is mean of determinations
7. Fig. 6. Define what is indicated by dotted line

8. line 183. Define $fT > MIC$
9. line 408. State how antimicrobials were administered.

Version 2:

Reviewer comments:

Reviewer #2

(Remarks to the Author)

Thank you for addressing my comments. I have no further concerns regarding this manuscript.

Reviewer #3

(Remarks to the Author)

General comment:

Toxicity is a principal concern of antimicrobials. Enthusiasm for the paper would be strengthened if toxicity to mammalian cells or animals was directly addressed in the manuscript.

RESPONSE: We thank Reviewer #3 for highlighting the strengths and weaknesses of the manuscript.

To address the principal concern of toxicity, we have included genotoxicity (human lymphocytes), mutagenicity (mouse lymphoma cells) and cytotoxicity (using human HepG2 cells) data, all of which were favourable for progression of the compound (lines 226 – 235).

Reviewer response: Enthusiasm for compound toxicity would be enhanced if more detail was presented including the data (not just a summary), dose, how many determinations, the positive and negative controls and refs for each toxicity test. In vivo data would also be helpful. The reviewer recommends toxicity as a separate section.

2. Supplementary Figure 2. In vitro activity of Debio 1453 against non-gonococcal Neisseria species.

Reviewer: Fig. S2. State data are a mean of X determinations +/- SD

RESPONSE: Data in Figure S2 are single determinations (i.e. not the mean) which is a limitation, however, this is mitigated by the testing of multiple different strains (n = 10) and multiple different concentrations (n = 4 per strain).

Reviewer response: Enthusiasm for MIC values would be enhanced with more than one determination.

5. Table 2. Frequency of selection for mutants with elevated minimum inhibitory concentration (MIC) 864 for three strains of Neisseria gonorrhoeae exposed to increasing concentrations of ciprofloxacin or 865 Debio 1453.

Reviewer: Table 2. State data are a mean of X determinations +/- SD...or a statement that values given are within a +/- percentage

RESPONSE: Data from Table 2 are not the mean of multiple determinations. Determining frequency of resistance is a resource intensive process, thus the testing of multiple different strains at various multiplicities of MIC was favoured over biological replication.

Reviewer response: A single data point is insufficient to make a conclusion regarding the frequency of resistance.

Reviewer #4

(Remarks to the Author)

As requested by the editor, I reviewed the authors response to the comment of Reviewer #1 on page 4, beginning with "Missing data relevant to the possible future of 1453 as a medicine..." through the authors response to page 5, which ends in the author response "We have added relevant early ADME information...".

• Missing data relevant to the possible future of 1453 as a medicine. The authors describe 1453 as “a promising developmental candidate for the treatment of gonorrhoea,” but this statement requires more substantiation.

RESPONSE: We agree that the use of the term “developmental candidate” in the abstract is premature given the data package that we propose to present in the current manuscript. Thus, we have deleted the term “development” and refer to it simply as a “candidate”. That being said, we have also now added PK, ADME and toxicology data.

With regard to the future of 1453 as a medicine, we are very pleased to inform the reviewers we recently secured renewed funding from CARB-X to complete our pre-clinical data package and will begin first in human studies in 2025.

REVIEWER RESPONSE:

I agree with the nuance made. The addition of this new data further adds on the proof that 1453 is a promising candidate for the treatment of gonorrhoea, and further improves the manuscript.

o Most glaring is the lack of PK or PK/PD information for 1453. The authors describe in some detail the benchmarking efforts regarding ceftriaxone PK and PK/PD as a form of validation of the model. While this is helpful, the subject of the manuscript is 1453. PK and PK/PD results for 1453 are also highly relevant.

RESPONSE: Single dose (80 mg/kg Debio 1453 equivalent dosed as Debio 1453P formulated in 5% dextrose) oral PK data generated from *N. gonorrhoeae* infected animals (n = 3 animals at each of 0.083, 0.25, 0.5, 1, 2, 4, 8, and 24-hours post-dose), has now been included in the manuscript with commentary in the results (Figure 6, line 236 - 244)

For the purpose of the current review, we feel it is important to inform the reviewers that we have generated a much broader PK/PD package for Debio 1453 including data from both the *N. gonorrhoeae* vaginal model (single dose PK at multiple oral doses in infected animals, dose fractionation for *N. gonorrhoeae* ATCC 700825, dose range efficacy for four strains [ATCC 700825, AR Bank0179-15, AR Bank0181-17, WHO X-07 presented in the current manuscript Fig. 7]) and the *S. aureus* neutropenic thigh model (single dose PK at multiple oral doses in *S. aureus* infected animals, dose ranging efficacy ATCC 29213 [current manuscript Fig. 8]). We generated popPK models for each data set (vaginal model and thigh model), assessed the driver for efficacy and defined PK/PD targets for efficacy. Pleasingly, the results from each model are in alignment; Debio 1453 PK/PD targets generated using the vaginal model that were associated with ceftriaxone efficacy (2- and 3log₁₀CFU/mL lavage; current manuscript Fig. S4) aligned with stasis and 1log₁₀CFU/gram reductions using the thigh model, which were targets used for the clinical development of recent anti-gonococcal zoliflodacin and gepatodacin. We believe that the extended PK/PD package described briefly above is beyond the scope of the current “foundational” manuscript (medicinal chemistry, biochemistry, structural studies, microbiology, and proof of concept in vivo efficacy), and warrants a subsequent, dedicated manuscript. Further, we intend for the subsequent PKPD manuscript to also include human PK data, which we will be collected in upcoming phase 1 studies (2025), thus precluding its submission in the immediate term.

REVIEWER RESPONSE:

I agree with the authors that including the extended PK/PD package for Debio 1453 should be described in a dedicated manuscript focusing on its PKPD characteristics only. The current manuscript, with the current additions, suffices as a foundational manuscript.

For Figure 6, I propose to indicate the lower limit for quantification in the graph.

o The manuscript would ideally include other relevant ADMET information as well, since the compound is positioned as a “developmental candidate.”

RESPONSE: We have added relevant early ADME information including mouse plasma protein binding, permeability in Caco-2 cells to inform bioavailability and metabolic stability using human HepG2 cells (lines 211 – 225).

REVIEWER RESPONSE:

Addition of these data further improves the manuscript.

Version 3:

Reviewer comments:

Reviewer #3

(Remarks to the Author)

I. Drug Toxicity

Three biological (independent) replicates are necessary to make a conclusion.

Table S2. State biological or technical replicate

Table S3. 2 biological replicates

Table S5. 2 replicates: state biological or technical

II. MIC determination (Fig. S2)

Response indicates “we performed two additional biological replicates, making a total of three. Fig. S2 has been updated using the mode value for each isolate.”

However, there are no indicated changes to the data in Fig. S2. Please clarify.

III. Table 2 Resistance legend and Results text (166-171).

1. As indicated earlier, a single data point is insufficient to make a conclusion regarding the frequency of resistance. The revised Table 2 has a single data point for 3 of 4 strains tested.

2. The revised legend indicates an inoculum of $\sim 10^8$ CFU for strain ATCC 700825, and $\sim 10^9$ CFU for the other 3 strains tested. However, the frequency of resistance is $< 10^9$ CFU and $< 10^{10}$ CFU, respectively. Please provide clarification on methods for resistance calculation.
3. State the resistance frequency range at the highest MIC challenge for the strains tested in Results text.

Reviewer #1 (Remarks to the Author):

This is a review provided for a manuscript entitled “The bactericidal FabI inhibitor Debio 1453 clears antibiotic-resistant *Neisseria gonorrhoeae* infection in vivo” from Gerusz, et al.

The authors describe an interesting small molecule inhibitor of Ng FabI, Debio 1453, with a range of data including biochemical, microbiological, and structural studies as well as a demonstration of in vivo efficacy in a relevant animal model of infection. The manuscript is generally well written, and the results suggest a positive trajectory for Debio 1453. Nevertheless, it is difficult to ascertain the significance to the field as a whole given uncertainty about the development status of 1453, a lack of information regarding safety and other properties, limited information about comparator studies relevant to innovation, and some questions about the methodology employed.

Notable strengths of the manuscript and of the compound include:

- Potent anti-gonorrheal activity (MICs) against both the WHO reference panel (Table 1) and against 100 recent clinical isolates (Table 3, MIC₉₀ 0.06 µg/mL). These results importantly also indicate a lack of cross-resistance with existing therapies, as would be expected based on the mechanism of action.
- A low resistance frequency validated in three different strains of Ng.
- Mechanism of action studies including biochemical assays (with potent, sub-nanomolar inhibition of Ng FabI), selection for mutations in *fabI* under drug pressure, and x-ray crystallography.
- An intriguing binding mode incorporating interactions with both the FabI enzyme target and the NADH cofactor. The authors hypothesize that the immutability of the cofactor contributes to the relatively low frequency of resistance.
- In vivo efficacy upon twice-daily oral dosing in the murine vaginal infection model using 4 different strains of Ng (Figure 5).
- Together, the results above show Debio 1453 to be a potent and efficacious lead compound with a validated mechanism of action and a low likelihood of rapid resistance selection.

RESPONSE: We thank Reviewer #1 for their careful and considerate review and for highlighting the strengths of the manuscript. We provide responses to address each of the weaknesses identified below.

Weaknesses of the manuscript include:

- Limited discussion of relevant background and historical results, making it more difficult to assess the level of field-shifting innovation.
 - o For example, the authors state that “the structures defined in this study are the first published NgFabI-inhibitor structures...” How does the binding mode observed/presented in this manuscript compare to previously published structures of FabI inhibitors with the enzyme from other bacteria? [for example the structure from reference 21 and that found in the important but uncited reference ACS Central Science 2022, 8, 1145]? Are the interactions fundamentally different? Does this have any implications for the authors’ hypothesis regarding resistance emergence?

RESPONSE: This is an interesting point that we have now addressed in the discussion (lines 292 – 301) with additional references (including ACS Central Science 2022, 8, 1145). In short, our binding mode follows the generally observed ternary complex “FabI + inhibitor + co-factor” also described in previously published structures. Moreover, our higher resolution structure helped the medicinal chemistry design by allowing us to leverage previously underappreciated interactions of our pyrido-diazepane substituents (e.g. the hydroxyl group in Debio 1453) with a precisely defined water network binding back to the co-factor, which improved enzymatic potency. This high-resolution co-structure also helped us to formulate our hypothesis that maximizing the interactions with the non-mutable co-factor may contribute to a lower resistance emergence. This hypothesis is then supported by the single mutant FabI allele associated with increased Debio 1453 MIC (L257F) being outside of the active site (additional discussion lines 307 - 312 and new Supplemental Figure 5). Key learnings related to SAR are also now included to further illustrate the advance (Supplementary Table 1).

o The authors mention “the previously described on-target inhibition of NgFabI by Debio 1452 (IC₅₀ ~100 nM)¹⁵”. The reviewer could not find this value in the cited work but did see a Kiapp of 5 nM. Given its important role in the genesis of the research described here, why wasn’t 1452 tested as a control in the current work using the same biochemical assay methodology? This is particularly striking since MIC results from 1452 are included in Table S1.

RESPONSE: The positive control used to set up the NgFabI assay in these studies was an earlier compound called DPM011876 (see below) that was structurally similar to Debio 1452, which showed an IC₅₀ for NgFabI of 534 nM. Since our lead optimization program started in 2017, we deduced the IC₅₀ ~100 nM of Debio 1452 from Figure 3A of reference ¹⁵ (published in 2016) but did not reassess its enzyme potency since its MIC₅₀ & MIC₉₀ were considered too high. These experiments were done a few years ago and the provider for this assay no longer operates. To avoid confusion, we have removed the discussion about Debio 1452 IC₅₀ from the results section.

o The authors state: “Until now, in vivo assessments for new anti-gonococcal drugs were not reported using *N. gonorrhoeae* animal models due mainly to spontaneous eradication of bacteria...” This statement is highly misleading, as several previously published studies include in vivo results from a murine vaginal infection model, including cited reference 22; Microb. Pathog. 2022, 164, 105454; J. Med. Chem. 2024, 67, 15537; and PLOS One 2022, 17, e0266764.

RESPONSE: Indeed, the original text on the use of the estradiol-treated mouse model was misleading and its inclusion was a mistake that occurred during internal review of the manuscript. It has now been removed and replaced by the following: “The estradiol-treated mouse model of vaginal colonisation is an emerging tool for the assessment of antimicrobial efficacy” with the addition of appropriate references including each of those mentioned by the reviewer (line 325 – 326).

Opportunities to enhance the discussion of the results.

o The authors obtained a mutant encoding an L257F amino acid substitution in FabI from strain 6804 but do not rationalize how this might lead to resistance based on their structural studies.

RESPONSE: We have added in the discussion a short paragraph addressing our single observed mutant (see response to next point) with an additional explanatory figure (new Supplementary Fig. 5, see lines 307 - 312). We provide a hypothesis rationalizing how this mutation, located at the interface of 2 FabI subunits found only in the tetrameric state, might lead to resistance by disrupting the oligomeric FabI transition. Also, the absence of any mutation found directly in the active site is strengthening support for our hypothesis that maximizing the interactions of Debio 1453 with the non-mutable co-factor might generate an inhibitor difficult to directly destabilize in the active site without a higher fitness-cost for enzyme functionality.

o Results from the other strains used in the resistance study are not presented.

RESPONSE: The frequency of single-step selection for reduced susceptibility data comes from a larger study that included a total of 9 different FabI-inhibitor compounds against the three strains mentioned in the current manuscript submission. Here, diverse mutations to *fabI* were identified resulting in various amino acid substitutions to the predicted protein sequence, which were associated with up to 8-fold increases in MIC to the specific inhibitor used for selection. However, only the two clones exposed to the Debio 1453 compound mentioned in the manuscript were stably propagated and revealed an increase in MIC of ≥ 2 -fold (i.e. one doubling dilution) for Debio 1453. Whilst colonies appearing upon Debio 1453 exposure in the assay were patched again onto agar plates containing the same concentration of Debio 1453 used for selection, this did not always correlate with a ≥ 2 -fold increase in MIC. For clarity, the methods section has been updated (lines 512-522) and terminology surrounding the results unified (by replacing 'selection for increased MIC' with 'selection for reduced in vitro susceptibility') throughout the text.

Opportunities for improved clarity.

o In presenting results from time-kill assays, the authors should specify in the text at which MIC multiples bactericidal activity was observed (not just in the figure legend).

RESPONSE: The text has been updated to include "bactericidal killing... achieved at all of the concentrations tested, including as low as 2X MIC", which reiterates the statement in the following sentence "The mean time to reach bactericidal activity... 10 hours at 2X MIC, 8.5 hours at 4X MIC, 8 hours at 8X MIC and 8 hours at 16X MIC" (lines 170 – 174).

o The authors don't state the resolution of the crystal structure in the text.

RESPONSE: The resolution (1.3 Å) is now included in the discussion (line 298) as well as in Supplementary Table 2.

- o Statistical significance between groups in Figure 5 is not addressed.

RESPONSE: Statistical significance has been assessed and is now included for Figure 7 and Figure 8 (originally Fig 5 and 6). The figure legends and methods have been updated accordingly.

- o In the *S. aureus* infection model, linezolid is dosed at 3x100 mg/kg “predicted to produce exposures higher than the human clinical equivalent (120 mg twice daily for 600 mg human dose).33”

- The wording above is confusing.
- 120 mg should be 120 mg/kg
- It appears to state that 2x120 mg/kg is equivalent to a 600 mg human dose of linezolid. Linezolid is dosed twice daily in humans at 600 mg per dose. 2x120 mg/kg in mice is equivalent to 2x600 mg in humans.
- Reference 33 is a secondary citation; the original source should be cited. (Antimicrob. Agents Chemother. 2011, 55, 3453).

RESPONSE: The error in the text has been rectified and now reads “120 mg/kg twice daily dosing in mice emulates 600 mg twice daily human dosing” and the appropriate reference has now been included (line 275).

- Missing data relevant to the possible future of 1453 as a medicine. The authors describe 1453 as “a promising developmental candidate for the treatment of gonorrhoea,” but this statement requires more substantiation.

RESPONSE: We agree that the use of the term “developmental candidate” in the abstract is premature given the data package that we propose to present in the current manuscript. Thus, we have deleted the term “development” and refer to it simply as a “candidate”. That being said, we have also now added PK, ADME and toxicology data.

With regard to the future of 1453 as a medicine, we are very pleased to inform the reviewers we recently secured renewed funding from CARB-X to complete our pre-clinical data package and will begin first in human studies in 2025.

- o Most glaring is the lack of PK or PK/PD information for 1453. The authors describe in some detail the benchmarking efforts regarding ceftriaxone PK and PK/PD as a form of validation of the model. While this is helpful, the subject of the manuscript is 1453. PK and PK/PD results for 1453 are also highly relevant.

RESPONSE: Single dose (80 mg/kg Debio 1453 equivalent dosed as Debio 1453P formulated in 5% dextrose) oral PK data generated from *N. gonorrhoeae* infected animals (n = 3 animals at each of 0.083, 0.25, 0.5, 1, 2, 4, 8, and 24-hours post-dose),

has now been included in the manuscript with commentary in the results (Figure 6, line 236 - 244)

For the purpose of the current review, we feel it is important to inform the reviewers that we have generated a much broader PK/PD package for Debio 1453 including data from both the *N. gonorrhoeae* vaginal model (single dose PK at multiple oral doses in infected animals, dose fractionation for *N. gonorrhoeae* ATCC 700825, dose range efficacy for four strains [ATCC 700825, AR Bank0179-15, AR Bank0181-17, WHO X-07 presented in the current manuscript Fig. 7]) and the *S. aureus* neutropenic thigh model (single dose PK at multiple oral doses in *S. aureus* infected animals, dose ranging efficacy ATCC 29213 [current manuscript Fig. 8]). We generated popPK models for each data set (vaginal model and thigh model), assessed the driver for efficacy and defined PK/PD targets for efficacy. Pleasingly, the results from each model are in alignment; Debio 1453 PK/PD targets generated using the vaginal model that were associated with ceftriaxone efficacy (2- and 3log₁₀CFU/mL lavage; current manuscript Fig. S4) aligned with stasis and 1log₁₀CFU/gram reductions using the thigh model, which were targets used for the clinical development of recent anti-gonococcal zoliflodacin and gepatodacin.

We believe that the extended PK/PD package described briefly above is beyond the scope of the current “foundational” manuscript (medicinal chemistry, biochemistry, structural studies, microbiology, and proof of concept in vivo efficacy), and warrants a subsequent, dedicated manuscript. Further, we intend for the subsequent PKPD manuscript to also include human PK data, which we will be collected in upcoming phase 1 studies (2025), thus precluding its submission in the immediate term.

o The manuscript would ideally include other relevant ADMET information as well, since the compound is positioned as a “developmental candidate.”

RESPONSE: We have added relevant early ADME information including mouse plasma protein binding, permeability in Caco-2 cells to inform bioavailability and metabolic stability using human HepG2 cells (lines 211 – 225).

• Missing data relevant to SAR.

o The authors mention results from >300 analogues. Unless a separate manuscript is planned (which would be acceptable), a supplementary table presenting these results would be highly useful.

RESPONSE: We have added additional analogues in Supplementary Table S1 that provide key elements of SAR for the series, we also discuss them briefly in the Results section (line 121 – 132).

o The authors describe the importance of the stereochemistry of substituents in driving both interactions with the target and potency. These claims cannot be properly evaluated without data from other stereoisomers.

RESPONSE: The data corresponding to the stereoisomers has been added in Table S2 and briefly discussed in the results section (line 126, line 130).

- Minor issues with characterization data.

- o For compound 2, a coupling constant of 20.7 Hz is reported as part of a 2H dtd. There are not 2 chemically equivalent protons that could give rise to a dtd, and a 20.7 Hz coupling constant is unusually large. A likely explanation is that there are two overlapping 1H peaks that need to be separately analyzed.

RESPONSE: Indeed, the reported coupling constant of 20.37 Hz was a mistake, probably arising from the fact that the amide rotamers have not been fully resolved at 90°C. We have rerun the NMR at this temperature and corrected the assignment, we also provide the spectrum at room temperature for comparative purposes (see lines 381 – 389).

- o In the two-step synthesis of Debio 1453P, step one affords 1.35 g of product, but step two begins with 3.45 g. Was step one conducted multiple times to afford the 3.45 g, or was a separate, larger-scale step one conducted?

RESPONSE: Yes, step one was conducted multiple times, which provided enough material for the larger step two reported synthesis. The text has been updated to reflect this (line 415).

Reviewer #2 (Remarks to the Author):

This is a well-written and thorough study on the optimisation of the FabI -inhibitor, Dehbio 1452. I do note that prior studies have used other estradiol-treated mouse models of vaginal colonisation to report efficacy of antimicrobials and indeed have reported Pk/Pd for ceftriaxone (See A. Jerse studies) (line 220). It would be worth commenting on why you felt that ovariectomies were necessary in an estradiol-treatment model.

RESPONSE: We appreciate the positive feedback from Reviewer #2.

Indeed, the original text on the use of the estradiol-treated mouse model was misleading and has now been removed and replaced by the following: “The estradiol-treated mouse model of vaginal colonisation is an emerging tool for the assessment of antimicrobial efficacy” with the addition of appropriate references (see response to reviewer #1 above).

Ovariectomies were performed to avoid the need to stage mice prior to administration of estradiol. This is now reflected in the text (line 577) with an appropriate reference.

Connolly, K. L. et al. Preclinical testing of vaccines and therapeutics for gonorrhea in female mouse models of lower and upper reproductive tract infection. *The Journal of Infectious Diseases* 224, S152-s160, doi:10.1093/infdis/jiab211 (2021)

I felt that adding a commentary on whether Dehbio would result in escape mutants in commensal *Neisseria* spp. would be worth adding here. It is well known that escape mutations appear preferentially in the non-clinical species and so showing that they are not under selective pressure would add to the case for continued development of this promising lead compound.

RESPONSE: This is an important suggestion. In addition to adding commentary, we chose to test the in vitro activity of Debio 1453 against 112 isolates from 16 non-gonococcal *Neisseria* species, i.e. 15 commensal *Neisseria* species and *N. meningitidis* of different genogroups. Debio 1453 showed differential activity against each species which correlated with the degree of conservation between each species FabI and NgFabI. Propensity for horizontal movement of *fabI* alleles from commensals to *N. gonorrhoeae* was assessed using public genome databases and no non-NgFabI enzymes were identified in *N. gonorrhoeae* (38,623 *N. gonorrhoeae* genomes assessed). See updated text (lines 154 – 164; lines 313 - 322) and Supplementary Figure 2.

Reviewer #3 (Remarks to the Author):

General:

N. gonorrhoea strains have developed resistance to nearly all medications and novel treatments are sorely needed.

Strengths:

The authors describe a new antimicrobial compound that inhibits an essential function involved in fatty acid synthesis in bacteria. The compound displayed antibacterial activity against a broad range of N. gonorrhoea strains; exhibited efficacy in a murine vaginal gonorrhoea model and S. aureus neutropenic murine thigh model; and showed very low bacterial resistance.

Weakness:

Toxicity is a principal concern of antimicrobials. Enthusiasm for the paper would be strengthened if toxicity to mammalian cells or animals was directly addressed in the manuscript.

RESPONSE: We thank Reviewer #3 for highlighting the strengths and weaknesses of the manuscript.

To address the principal concern of toxicity, we have included genotoxicity (human lymphocytes), mutagenicity (mouse lymphoma cells) and cytotoxicity (using human HepG2 cells) data, all of which were favourable for progression of the compound (lines 226 – 235).

Specific

1. Table 1. Define what is depicted by shading

RESPONSE: Shading depicts resistance. Indeed, in the original submission several instances of resistance were not highlighted appropriately. Upon reviewing Nature Communications submission guidelines, we have now removed the shading.

2. Fig. S2. State data are a mean of X determinations +/- SD

RESPONSE: Data in Figure S2 are single determinations (i.e. not the mean) which is a limitation, however, this is mitigated by the testing of multiple different strains (n = 10) and multiple different concentrations (n = 4 per strain).

3. Fig. 3. Define what is indicated by dotted line

RESPONSE: The Figure 3 legend has been updated to include: “The dotted line indicates 90% of cumulative isolates (MIC90).”

4. Fig. 4B/C / Fig. S3. Add the type of cultured mammalian cells used for intracellular killing assays to legend; consider merging the figures in manuscript so controls are shown with the experiment.

RESPONSE: Data from Figure 4B/C and Figure S3 have been merged to form the new Figure 5 so that azithromycin control data can be easily read alongside Debio 1453 data.

The type of cultured mammalian cells used for intracellular assays has now been included in the title of the new Figure 5 legend (“Intracellular killing of *Neisseria gonorrhoeae* in cultured HeLa229 human cervix carcinoma cells.”)

5. Table 2. State data are a mean of X determinations +/- SD...or a statement that values given are within a +/- percentage

RESPONSE: Data from Table 2 are not the mean of multiple determinations. Determining frequency of resistance is a resource intensive process, thus the testing of multiple different strains at various multiplicities of MIC was favoured over biological replication.

6. Fig. 5. Define horizontal bar is mean of determinations

RESPONSE: The figure legend of Figs 7 and 8 have been updated to include: “Data for each group is summarised using the mean (black horizontal bars).”

7. Fig. 6. Define what is indicated by dotted line

RESPONSE: The Figure 8 legend has been updated to include: “The dotted grey lines depict the mean CFU/mL at the start of treatment.”

8. line 183. Define $fT > MIC$

RESPONSE: The sentence has been updated to include “... time of free concentration above MIC ($fT > MIC$)”. Line 261

9. line 408. State how antimicrobials were administered.

RESPONSE: The sentence has been updated to include “...via intraperitoneal (IP) injection...” line 597 – 598.

Reviewer #2 (Remarks to the Author):

Thank you for addressing my comments. I have no further concerns regarding this manuscript.

AUTHOR RESPONSE: We thank reviewer #2 for helping us to improve the manuscript.

Reviewer #3 (Remarks to the Author):

Reviewer response: Enthusiasm for compound toxicity would be enhanced if more detail was presented including the data (not just a summary), dose, how many determinations, the positive and negative controls and refs for each toxicity test. In vivo data would also be helpful. The reviewer recommends toxicity as a separate section.

AUTHOR RESPONSE: We have updated the supplemental material to include the toxicity data including dose, number of determinations and controls (Supplementary Tables 2, 3, 4, 5, 6). Refs for each toxicity test are provided in the methods. Toxicity is now a separate section in the Results.

Supplementary Figure 2. In vitro activity of Debio 1453 against non-gonococcal Neisseria species.

Enthusiasm for MIC values would be enhanced with more than one determination.

AUTHOR RESPONSE: In response to the comment, we performed two additional biological replicates, making a total of three. Fig. S2 has been updated using the mode value for each isolate. The number of biological replicates is now stated in the figure legend. The text discussing these results has been modified (line 133-139).

A single data point is insufficient to make a conclusion regarding the frequency of resistance.

AUTHOR RESPONSE: In response to the comment, we performed an additional frequency of resistance assessment for *N. gonorrhoeae* strain ATCC 700825 which was performed in biological triplicate. All data is presented in Table 2 (18 data points) and discussed in the text (line 166 – 176).

Reviewer #4 (Remarks to the Author):

I agree with the nuance made [referring to future of Debio 1453 as a medicine]. The addition of this new data further adds on the proof that 1453 is a promising candidate for the treatment of gonorrhea, and further improves the manuscript.

I agree with the authors that including the extended PK/PD package for Debio 1453 should be described in a dedicated manuscript focusing on its PKPD characteristics only. The current manuscript, with the current additions, suffices as a foundational manuscript.

Addition of these data [ADME] further improves the manuscript.

AUTHOR RESPONSE: We thank reviewer #4 for supporting our approach to address the comments of Reviewer #1.

For Figure 6, I propose to indicate the lower limit for quantification in the graph.

AUTHOR RESPONSE: We have considered the suggestion, however, we believe that it may confuse the interpretation of the figure. The free fraction is a calculation based off total drug (which is what was quantified) and thus many of the calculated values fall below the limit of quantification (of total drug). We have chosen to describe the lower limit for quantification in the legend of the figure.

REVIEWERS' COMMENTS

Reviewer #3 (Remarks to the Author):

I. Drug Toxicity

Three biological (independent) replicates are necessary to make a conclusion.

Table S2. State biological or technical replicate

RESPONSE: Biological is now stated below the table.

Table S3. 2 biological replicates

RESPONSE: Biological is stated below the table.

Table S5. 2 replicates: state biological or technical

RESPONSE: Biological is now stated below the table.

II. MIC determination (Fig. S2)

Response indicates “we performed two additional biological replicates, making a total of three. Fig. S2 has been updated using the mode value for each isolate.”

However, there are no indicated changes to the data in Fig. S2. Please clarify.

RESPONSE: Fig. S2 was updated in the most recent submission. All data including each triplicate is available in the Source Data File.

III. Table 2 Resistance legend and Results text (166-171).

1. As indicated earlier, a single data point is insufficient to make a conclusion regarding the frequency of resistance. The revised Table 2 has a single data point for 3 of 4 strains tested.

2. The revised legend indicates an inoculum of $\sim 10^{+8}$ CFU for strain ATCC 700825, and $\sim 10^{+9}$ CFU for the other 3 strains tested. However, the frequency of resistance is $< 10^{+9}$ CFU and $< 10^{+10}$ CFU, respectively. Please provide clarification on methods for resistance calculation.

RESPONSE: Frequency of resistance calculation is provided in the methods section. “Frequencies were generated by dividing the number of stable colonies with reduced Debio 1453 susceptibility by the inoculated CFU.” In the table description, the limit of detection is mentioned (one colony). The inoculum for ATCC 700825, for example, was 1.57×10^8 . Following the calculation in the methods ($1 / 1.57 \times 10^8$) equals 6.38×10^{-9} . When no colony was identified, it was reported as $< 6.38 \times 10^{-9}$ (as stated in the Table footnote). All data is available in the Source Data File.

3. State the resistance frequency range at the highest MIC challenge for the strains tested in Results text.

RESPONSE: Resistance frequency at the highest MIC challenge is now stated in the results (line 169 and 173-174).